# Robust Mendelian randomization in the presence of residual population stratification, batch effects and horizontal pleiotropy

Carlos Cinelli[1]✉, Nathan LaPierre[2], Brian L. Hill[2], Sriram Sankararaman[2,3,4] & Eleazar Eskin[2,3,4]

Mendelian Randomization (MR) studies are threatened by population stratification, batch effects, and horizontal pleiotropy. Although a variety of methods have been proposed to mitigate those problems, residual biases may still remain, leading to highly statistically significant false positives in large databases. Here we describe a suite of sensitivity analysis tools that enables investigators to quantify the robustness of their findings against such validity threats. Specifically, we propose the routine reporting of sensitivity statistics that reveal the minimal strength of violations necessary to explain away the MR results. We further provide intuitive displays of the robustness of the MR estimate to any degree of violation, and formal bounds on the worst-case bias caused by violations multiple times stronger than observed variables. We demonstrate how these tools can aid researchers in distinguishing robust from fragile findings by examining the effect of body mass index on diastolic blood pressure and Townsend deprivation index.

[1] Department of Statistics, University of Washington, Seattle, WA, USA. [2] Department of Computer Science, University of California, Los Angeles, CA, USA. [3] Department of Human Genetics, University of California, Los Angeles, CA, USA. [4] Department of Computational Medicine, University of California, Los Angeles, CA, USA. ✉email: cinelli@uw.edu

Many fundamental questions in the social and medical sciences are questions of cause and effect. For instance, what are the social and health consequences of obesity? In practice, however, it is often infeasible or unethical to perform a randomized controlled trial to answer these types of questions. Moreover, observational studies are prone to being biased due to the presence of unmeasured confounders. In such cases, the method of instrumental variables[1–4] (IVs) may be an appealing alternative, allowing one to infer cause-effect relationships even in the presence of unmeasured confounding between the exposure and the outcome.

Mendelian randomization (MR) exploits genetic variants associated with an "exposure" trait of interest as IVs to investigate whether that exposure has a causal effect on an "outcome" trait of interest[5–11]. The technique of MR has become a standard tool for inferring causal relationships, with numerous applications published in medical, genetic and epidemiological journals[6–14]. This growth has been accelerated by the availability of large genetic databases[15] and Genome-Wide Association Studies (GWAS) linking many genetic variants to complex phenotypes[8]. Nevertheless, the validity of MR studies depends on its own set of assumptions, and this rapid growth has not been accompanied with sufficient attention to those assumptions[16–23].

In particular, beyond being associated with the exposure, for a genetic variant to be a valid IV it must satisfy two important and often (though not always[24–28]) untestable conditions[6–9]: (i) it must not be itself confounded with the outcome trait; and, (ii) it must affect the outcome trait only through its effect on the exposure trait. These conditions may be violated in several ways due to populational and methodological artifacts, as well as biological mechanisms. Most notably, population stratification[29–34] and batch effects[34–36] are well-known sources of confounding biases in high-throughput genomic data. Likewise, many genetic variants tend to exert horizontal pleiotropy, meaning they affect the outcome trait through channels other than the exposure trait[37,38].

The prevailing method for dealing with population stratification and batch effects in MR is to adjust for genomic principal components and surrogate technical covariates representing genomic batch or assessment center[20]. In the case of horizontal pleiotropy, researchers are advised to perform alternative analyses that rely on modified identification assumptions (such as MR-Egger[39], MR-PRESSO[40], MR-MBE[41], MR-Mix[42], MR-GENIUS[43], among a plethora of variations). Although these methods have proved useful for partially mitigating these problems, residual biases may still remain[9,44]. Since those biases are impervious to sample size, they may lead to highly statistically significant false findings with large genomic data, as we demonstrate later via simulations.

Here we build on recent developments of the sensitivity analysis literature in statistics[45–50] to provide a suite of sensitivity analysis tools for MR studies that quantifies the robustness of inferences to the presence of residual population stratification, batch effects, and horizontal pleiotropy. Specifically, we introduce robustness values[46] (RV) for MR, summarizing the minimal strength that residual biases must have (in terms of variance explained of the genetic instrument and of the phenotypes) in order to explain away the MR causal effect estimate. We also provide intuitive sensitivity plots that allow researchers to quickly inspect how their inferences would have changed under biases of any postulated strength. Finally, we show how to place formal bounds on the worst-case bias caused by putative unmeasured variables with strength expressed in terms of multiples of the effect of observed variables, thereby facilitating expert judgment regarding the plausibility of such strong violations of the traditional MR assumptions. We demonstrate how these tools can aid researchers in distinguishing robust from fragile findings by examining the sensitivity of the effect of body mass index on diastolic blood pressure and Townsend deprivation index.

## Results

**MR-SENSEMAKR overview—a suite of sensitivity analysis tools for MR.** We developed MR-SENSEMAKR[51], a suite of sensitivity analysis tools for MR that allows researchers to perform robust inferences of causal effect estimates in the presence of violations of the standard MR assumptions. These tools quantify both how much the inferences would have changed under a postulated degree of violation, as well as the minimal strength of violation necessary to overturn a certain conclusion. MR-SENSEMAKR builds on an extension of the "omitted variable bias" framework for regression analysis[46,47] to the Anderson–Rubin method[52] and Fieller's theorem[53] for testing null hypotheses in the IV setting. This approach has a number of benefits, such as: (i) correct test size regardless of instrument strength; (ii) handling multiple confounding or pleiotropic effects acting simultaneously, possibly non-linearly; (iii) providing simple sensitivity statistics for routine reporting; and, (iv) exploiting expert knowledge to bound the maximum strength of biases (see Methods for details).

Let $D$ denote the "exposure" trait, $Y$ the "outcome" trait, and $Z$ the genetic instrument (e.g, a polygenic risk score). Additionally, let $X$ denote a set of observed "control" covariates which accounts for potential violations of the MR assumptions, such as population stratification (e.g, genetic principal components), batch effects (e.g, batch indicators) and traits that could block putative horizontal pleiotropic pathways[20]. Traditional MR analysis assumes that $X$ is sufficient for making $Z$ a valid instrumental variable for identifying the effect of the exposure trait $D$ on the outcome trait $Y$. An example for which this is the case is depicted in the directed acyclic graph (DAG) of Fig. 1a—in this example there are no pleiotropic pathways, and although there is confounding due to population structure, adjusting for $X$ (say, genomic principal components and batch indicators) is sufficient for eliminating all biases (note this DAG is for illustration purposes, as there are many alternative structures compatible with the IV assumptions; see Supplementary Information for other examples).

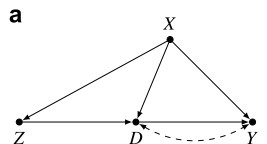 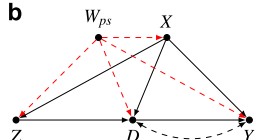 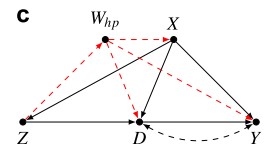

**Fig. 1 Directed acyclic graphs (DAGs) illustrating the traditional MR assumptions and possible violations.** Graphically, conditional on $X$, the genetic instrument $Z$ is a valid IV for the causal effect of trait $D$ on trait $Y$, if $X$ blocks all paths from $Z$ to $Y$ on the graph where the edge $D \rightarrow Y$ is removed[4]. **a** Example in which the IV conditions hold, and $X$ alone accounts for all population structure. **b** Here $X$ does not account for all population structure, and valid MR requires conditioning on both $X$ and $W_{ps}$. **c** Similarly, here we have a violation of the standard MR assumptions due to horizontal pleiotropy through trait $W_{hp}$; again, valid MR requires conditioning on both $X$ and $W_{hp}$.

The problem arises, however, when $X$ does not suffice for making $Z$ a valid instrument; instead, an extended set of control covariates would be necessary to do so, but some of these variables are, unfortunately, unobserved. Figure 1b and c illustrate two of such cases. In Fig. 1b, although X accounts for part of the confounding biases due to population structure (ps), it cannot account for all of it, and further adjustment for $W_{ps}$ would be necessary for making $Z$ a valid instrument. In Fig. 1c, we have a different type of problem; there, the genetic instrument exerts horizontal pleiotropy (hp) through trait $W_{hp}$, which needs to be accounted for in a valid MR analysis. In practice, of course, all these residual biases will often be acting simultaneously—we denote by $W$ the set of all additional unmeasured variables that would be necessary for making $Z$ a valid genetic instrument for the target effect of interest. If $Z$ is a valid instrument conditional on both $X$ and $W$, the target of estimation of traditional MR, denoted here by $\tau$, is the IV estimand consisting of a ratio of the two genetic associations

$$\tau = \frac{\beta_{YZ|XW}}{\beta_{DZ|XW}} \qquad (1)$$

where $\beta_{YZ|XW}$ is the partial regression coefficient of the genetic instrument with the outcome trait, and $\beta_{DZ|XW}$ the partial regression coefficient of the genetic instrument with the exposure trait. The interpretation of $\tau$ as a (weighed average of local) treatment effect(s) depends on additional functional constraints[3,26,54,55], but this has no consequence for our analysis (see Supplementary Information). Here we consider the case in which investigators are interested in the IV estimand $\tau$.

In this setting, MR-SENSEMAKR answers the following question: how strong would the unmeasured variables $W$ have to be such that, if accounted for in the analysis, they would have changed the conclusions of the MR study? As it has been extensively discussed elsewhere[7,9,17,20], MR studies are more reliable to test the presence or direction of a causal effect, rather than to precisely estimate its magnitude. Thus, here we focus on two problematic changes that $W$ could cause—turning a statistically significant result into an insignificant one; or, leading to unbounded or uninformative confidence intervals due to weak instruments (when using Fieller's theorem, confidence intervals can be: (i) connected and finite; (ii) the union of two disjoint unbounded intervals; or, (iii) the whole real line; see Methods).

It can be shown that, given a significance level $\alpha$, the confidence interval for the MR causal effect is unbounded if, and only if, we cannot reject the hypothesis that the genetic association with the exposure, $\beta_{DZ|XW}$, is zero. Likewise, the MR causal effect estimate is statistically insignificant if, and only if, we cannot reject the hypothesis that the genetic association with the outcome, $\beta_{YZ|XW}$ is zero (to understand this intuitively, note again that the MR estimate is the ratio of the genetic association with the outcome over the genetic association of the exposure. Note this ratio is zero if the numerator is zero; likewise, the ratio can be made arbitrarily large if the denominator can be made arbitrarily close to zero). Therefore, the problem of sensitivity analysis of the MR estimate can be reduced to the simpler problem of sensitivity analysis of these two genetic associations, and we can leverage all recent developments of sensitivity analysis for regression estimates (Cinelli and Hazlett[46,47,50]) to the sensitivity of MR.

MR-SENSEMAKR thus performs sensitivity analysis for the MR causal effect estimate by examining how strong $W$ needs to be to explain away either the observed genetic association with the exposure or the observed genetic association with the outcome. It deploys two main tools for assessing the sensitivity of these quantities. First, it computes key sensitivity statistics suited for routine reporting[46], including

- The partial $R^2$ of the genetic instrument with the (exposure/outcome) trait, revealing the minimal share of residual variation that $W$ needs to explain of the genetic instrument in order to fully eliminate the genetic association with the (exposure/outcome) trait;
- The robustness value ($RV_\alpha$) of the genetic instrument with the (exposure/outcome) trait, revealing the minimal share of residual variation (partial $R^2$), both of the genetic instrument and of the trait, that $W$ needs to explain in order to make the genetic association with the (exposure/outcome) trait statistically insignificant at the $\alpha$ level; and,
- Bounds on the maximum residual variation explained by unmeasured variables $W$ if they were as strong as: (i) observed principal components; (ii) measured batch effects; and, (iii) observed pleiotropic pathways.

MR-SENSEMAKR also provides sensitivity contour plots[46] that, given any hypothetical strength of $W$ (measured in terms of the partial $R^2$ of $W$ with the genetic instrument and with the trait), allows researchers to investigate what would have been the result of a significance test of the genetic association with the (exposure/outcome) trait had a $W$ with such strength been incorporated in the analysis (see Fig. 2). The sensitivity statistics can in fact be interpreted visually as summaries of critical lines of the contour plot. For instance, $RV_\alpha$ is the point of equal association of the critical contour corresponding to the significance level $\alpha$, whereas the partial $R^2$ corresponds to a vertical line tangent to the critical contour of zero, which is never crossed (see Methods for further details). Finally, these plots can also include several bounds on the maximum amount of residual variation that $W$ could explain, both of the genetic instrument and of the (exposure/outcome) trait, if $W$ were multiple times stronger than observed variables. Next, we apply these tools in a real example that examines the robustness of previous MR findings regarding the causal effect of BMI on blood pressure and deprivation[56–58].

**MR-SENSEMAKR helps distinguishing robust from fragile findings.** Previous studies[56–58] used MR on the UK Biobank data[15] to assess the causal effect of body mass index (BMI) on multiple outcome traits of interest. These MR analyses found a statistically significant effect of BMI on diastolic blood pressure (DBP)[57] and on Townsend deprivation index (deprivation)—a measure of socioeconomic status[56]. Following these studies, we filtered the data to only include people with self-reported white British ancestry who were not closely related, leaving a sample size of 291,274 people; the genetic instrument consisted of a polygenic risk score (PRS) derived from 97 SNPs previously found to be associated with BMI, with external weights given by the effect sizes from the GIANT study[57,59] (see Methods for details).

The first part of Table 1 reports the results of the traditional MR analysis of the effects of BMI both on DBP and on deprivation. As it is usually recommended[20] and following the original studies, these MR analyses further adjust for: age, gender, 20 leading genomic principal components, assessment center, batch indicators, as well as smoking and drinking status (both are putative pleiotropic pathways, especially for DBP[60–65]). In consonance with the previous studies, we found that the conventional MR analyses led to positive and statistically significant effects of BMI on both traits, at the 5% significance level. The results, however, rely on the assumptions of zero residual population stratification, zero batch effects and zero horizontal pleiotropy, which are unlikely to hold. We thus used MR-SENSEMAKR to investigate the robustness of these findings to potential violations of the standard MR assumptions.

We first examined the robustness of the genetic association with the exposure trait (BMI). Recall that, if the MR violations are

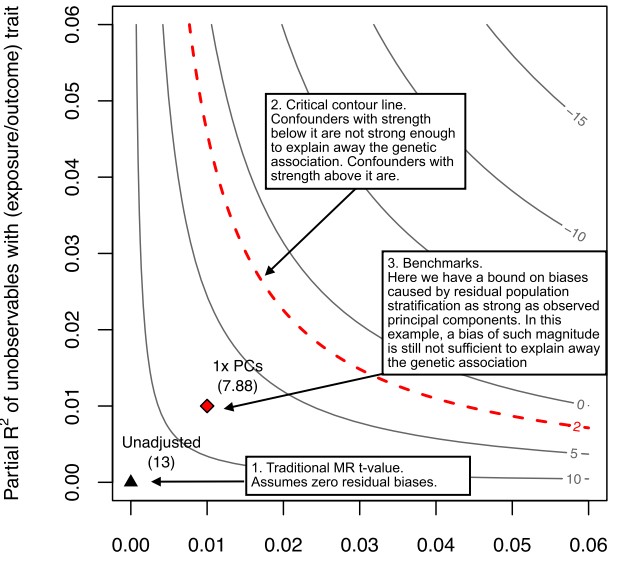

**Fig. 2 Sensitivity contour plot with benchmark bounds.** The horizontal axis shows the partial $R^2$ of unobserved variables $W$ with the genetic instrument; this corresponds to the percentage of residual variation of the genetic instrument explained by $W$. The vertical axis shows the partial $R^2$ of $W$ with the trait of interest, which can be either the exposure trait or the outcome trait; again, this stands for the percentage of residual phenotypic variance explained by $W$. Given any pair of partial $R^2$ values, the contour lines show the t-value that one would have obtained for testing the significance of the genetic association with the (exposure/outcome) trait, had a $W$ with such strengths been included in the analysis. The point represented by a black triangle (left lower corner) shows the t-value of a traditional MR study (i.e., $t = 13$)—note it assumes exactly zero biases due to unobserved variables $W$. As we move along both axes, the biases due to $W$ get worse, and can eventually be strong enough to reduce the t-value below a chosen critical level $t^*$, shown in the red dashed line (e.g., $t^* \approx 2$ for a significance level of $\alpha = 5\%$). Unobserved variables $W$ with strength below the critical red line are not strong enough to change the conclusions of the original MR study; on the other hand, unobserved variables $W$ with strength above the critical red line are strong enough to be problematic. The point represented by a red diamond bounds the maximum strength of $W$ if it were as strong as observed genomic principal components (1× PCs). They show the maximum bias caused by residual population stratification, if it had the same explanatory power as the PCs in explaining genetic and phenotypic variation. In this example, the plot reveals that residual population stratification as strong as the first genomic principal components would not be sufficient to make the genetic association statistically insignificant (i.e., the adjusted t-value accounting for a $W$ with such strength is 7.88, which is still above the critical threshold of $t^* \approx 2$). Finally, we note that if the unobserved variable $W$ is a singleton, then all the sensitivity analysis results are exact. If $W$ consists of multiple variables, then all sensitivity analysis results are conservative, meaning that this is the worst bias that a multivariate $W$ could cause if it had such strengths[46].

strong enough to explain away the genetic association with the exposure, this can lead to unbounded or uninformative confidence intervals for the MR causal effect estimate—the exercise we are performing here is thus tantamount to assessing the "weak instrument" problem, except that now we are accounting both for sampling uncertainty and potential violations due to unmeasured variables $W$. The results are shown in the section entitled "Sensitivity PRS-Exposure" of Table 1. (Note the results are the same both for DBP and deprivation, since the exposure trait, BMI, is the same in both cases.) The first

sensitivity measure is the partial $R^2$ of the PRS with BMI, which amounted to 1.67%. Although this quantity is already reported as a measure of instrument strength in many MR studies[20], it is perhaps less known that it is also a measure of its robustness to extreme confounding. In particular, this means that, even if the unmeasured variables $W$ explained all left-out variation in BMI, they would still need to account for at least 1.67% of the variance of the genetic instrument, otherwise $W$ cannot explain away the genetic association with the exposure. Next we obtained a robustness value of 11.88% for the PRS-exposure association. This means that any unmeasured variables $W$ that explain less than 11.88% of the residual variation, both of the PRS and of BMI, are not strong enough to make the genetic association with the exposure statistically insignificant.

Next we examined the robustness of the genetic association with the outcome traits; recall that any unobserved variables capable of explaining away the genetic association with the outcome trait are also capable of explaining away the MR causal effect estimate. The results are shown in the section entitled "Sensitivity PRS-Outcome" of Table 1, and here we have two separate results for each trait. Specifically, we obtained a partial $R^2$ of the PRS with DBP of 0.035% and a robustness value 1.47%. This means that, even if unobserved variables explained all variation of DBP, they still need to explain at least 0.035% of the residual variation of the genetic instrument to fully account for the observed PRS-DBP association; moreover, the RV reveals that unobserved variables that explain less than 1.47% of the residual variation, both of the genetic instrument and of DBP, are not sufficiently strong to overturn the statistical significance found in the original MR study. Moving to the next trait, the bottom row of Table 1 shows the sensitivity statistics for the effect of BMI on deprivation. Here we found a partial $R^2$ of 0.002% and a robustness value of 0.08%, revealing that much weaker residual biases would be able to overturn the MR effect estimate of BMI on deprivation.

Confronted with those results, the next step is to make plausibility judgments on whether unobserved variables with the strengths revealed to be problematic can be ruled out. To aid in these plausibility judgments, MR-SENSEMAKR computes bounds on the amount of variance explained by the unmeasured variables $W$ if it were as strong as observed variables. For our running example, these bounds are shown in Table 2; they reveal the maximum partial $R^2$ of unobserved variables $W$ with the genetic IV and with the traits, if it were as strong as: (i) 20 leading genomic principal components (1× PCs); (ii) observed batch and center effects (1× Batch+Center); and, finally, (iii) smoking and drinking status (1× Alc.+Smok.).

Starting with instrument strength, first note that all bounds on the PRS and BMI columns of Table 2 are (substantially) lower than than the RV of 11.88% for the genetic association with BMI; this means that, even if $W$ were as strong as those variables, this would not be sufficient to result in a "weak instrument" problem. Moreover, since all values of the PRS column are less than the partial $R^2$ of 1.67% of the variant-exposure association, even a "worst-case" $W$ that explains 100% of the variance of BMI, and as strongly associated with the genetic instrument as the observed variables, cannot account for the observed association of the genetic instrument with the exposure. Moving to statistical significance concerns, similar results hold for the PRS-DBP association. Since the bounds on both columns, for the PRS (column 1) and DBP (column 3), are below the robustness value of 1.47%, Table 2 reveals that biases as strong as the observed variables are not sufficient to make the MR causal effect estimate of BMI on DBP statistically insignificant. However, in stark contrast, note that all bounds on the PRS and deprivation columns are above the RV of 0.08% for deprivation, meaning that

**Table 1 Traditional MR results and sensitivity analyses.**

| | Traditional MR | | Sensitivity PRS-outcome | | Sensitivity PRS-exposure | |
|---|---|---|---|---|---|---|
| Outcome | Risk difference (95% CI) | P value | Partial $R^2$ | $RV_{\alpha = 0.05}$ | Partial $R^2$ | $RV_{\alpha = 0.05}$ |
| DBP | 0.145 (0.116–0.173) | $4.2 \times 10^{-22}$ | 0.035% | 1.47% | 1.67% | 11.88% |
| Deprivation | 0.033 (0.006–0.060) | 0.017 | 0.002% | 0.08% | | |

P-values correspond to two-sided t-tests in a two-stage least squares regression. No multiple testing corrections were performed.
CI Confidence Interval, DBP Diastolic Blood Pressure, MR Mendelian Randomization, PRS Polygenic Risk Score, RV Robustness Value.

**Table 2 Bounds on the maximum explanatory power of W (partial $R^2$), if it were as strong as: (i) 20 leading genomic principal components (1× PCs); (ii) observed batch and center (1× Batch+Center); and, (iii) smoking and drinking status (1× Alc.+Smok.).**

| | Bound partial $R^2$ with genetic IV | Bound partial $R^2$ with trait | | |
|---|---|---|---|---|
| W as strong as | PRS (Genetic IV) | BMI (Exposure) | DBP (Outcome) | Deprivation (Outcome) |
| 1× PCs | 0.20% | 0.11% | 0.05% | 0.37% |
| 1× Batch+Center | 0.06% | 0.07% | 0.84% | 15.76% |
| 1× Alc.+Smok. | 0.10% | 2.97% | 0.34% | 4.50% |

BMI Body Mass Index, DBP Diastolic Blood Pressure, IV Instrumental Variable, MR Mendelian Randomization, PCs Principal Components, RV Robustness Value.

unobserved variables **W** strong as those could easily overturn the original MR analysis.

Table 1 forms our proposed minimal reporting for sensitivity analysis in MR studies. Often, when supplemented with bounds such as those of Table 2, these metrics are sufficient to give a broad picture of the robustness of MR findings, as demonstrated above. Researchers, however, can refine their analyses and fully explore the whole range of robustness of their inferences with sensitivity contour plots, placing several different bounds on the strength of confounding multiple times stronger than observed variables.

The plots for DBP and deprivation are shown in Fig. 3 (see caption of Fig. 2 for details on how to read the plot). For DBP, note that neither residual population stratification up to 14× stronger than observed principal components nor residual batch-effects up to 6× stronger than observed batch-effects are sufficient to make the MR estimate statistically insignificant. Likewise, if residual pleiotropy were up to 7× stronger than important observed pleiotropic pathways, such as alcohol and smoking, this is also not sufficient to change the original conclusions. Finally, even if unobserved variables **W** had the same explanatory power of all the observed variables combined, this again would not change the results for DBP. In contrast, the sensitivity plot for deprivation reveals that the MR causal effect estimate of BMI on deprivation is sensitive to confounding with explanatory power as weak as a fraction (e.g, 0.5) of current observed variables. For completeness, Fig. 3c also shows the contour plots for the sensitivity of the genetic association with the exposure. We can see that, as already had been suggested by the RV of 11.88% and the bounds of Table 2, the PRS-BMI association is robust to relatively strong residual biases.

Putting these results in context requires assessing the quality of the benchmarks involved. For example, it does not seem unreasonable to argue that genomic principal components (PCs) correct for most, or at least a large part, of population structure[31], and that it is thus implausible to imagine residual population stratification multiple times stronger than what has been already corrected by observed principal components. Therefore, observed PCs may be useful benchmarks against this type of residual confounding. On the other hand, PCs may not be appropriate benchmarks against pleiotropy. Benchmarks for

horizontal pleiotropy require specific knowledge of the etiology of the disease, or of the social process under investigation. That is, researchers should inquire the main channels through which genetic effects could affect the outcome other than through the exposure. In this application, for instance, alcohol consumption is indeed suspected to be an important channel for horizontal pleiotropy in the case of DBP[60,61], and smoking also leads to a short-term increase in blood pressure (although its long-term effects are disputed)[62–64]. Therefore, one could plausibly argue that it is unlikely (although not impossible) that residual horizontal pleiotropy multiple times as strong as those still remains.

As to deprivation, the analysis reveals a more fragile finding. First, there is ambiguity as to the role of variables such as alcohol and smoking: they could be acting as proxies of pleiotropic pathways (genetics → behavior → deprivation), or acting as colliders, as it is plausible that greater deprivation causes increased consumption of alcohol and smoking. This is an unfortunate practical problem that can only be solved with better longitudinal data—if the measured smoking and alcohol consumption referred unambiguously to past behavior (relative to deprivation), then we could be assured they are not affected by the outcome. Given this ambiguity, in practice we recommend researchers still assess whether results are sensitive to such variables. In our example, not only there is no a priori reason to suspect that alcohol and smoking should be among the strongest pleiotropic pathways for deprivation, but the bounding exercise shows that residual pleiotropy a fraction as strong as those could easily overturn the MR results.

Overall, the sensitivity analyses suggest that: (i) the genetic association of the instrument (PRS) with the exposure (BMI) is relatively robust, and instrument strength is unlikely to be an issue; (ii) it would take substantial residual confounding and pleiotropy to reverse the original MR finding of the causal effect of BMI on DBP; and that, in contrast, (iii) the previous MR causal effect estimate of BMI on deprivation is fragile, meaning that there is little room for small residual biases, which could easily overturn the original analysis.

In the supplementary note we apply MR-SENSEMAKR to the MR analysis of the effects of High Density Lipoprotein (HDL) and Low Density Lipoprotein (LDL) on Coronary Artery Disease

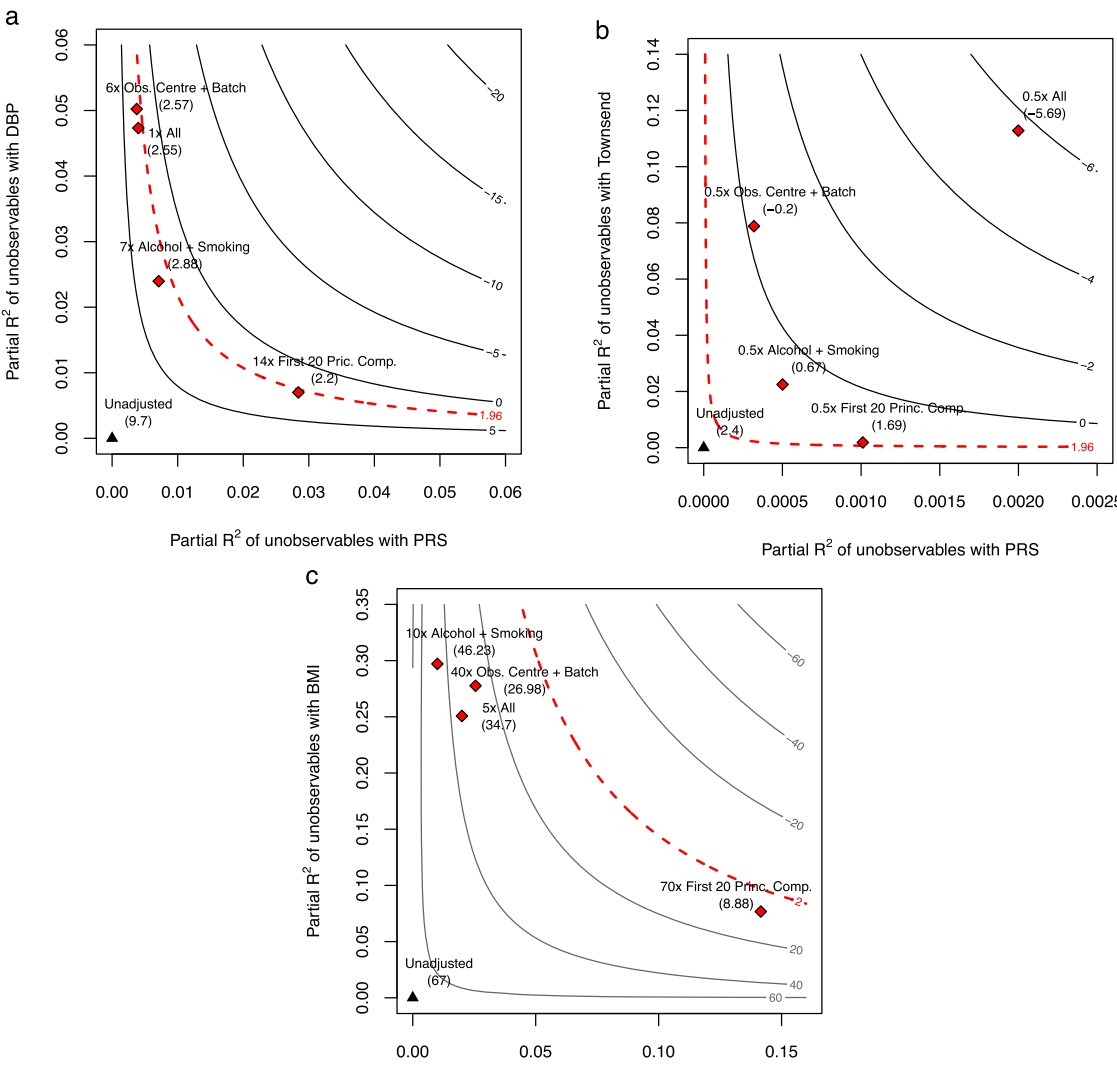

**Fig. 3 Sensitivity contours for the null hypothesis of zero effect of each genetic association. a** Diastolic Blood Pressure (DBP). **b** Townsend deprivation index (deprivation). **c** Body Mass Index (BMI). The caption of Fig. 2 provides details on how to interpret the contour plots.

(CAD). We find that, although the strength of association is similar for both exposure traits, the effect of LDL on CAD is relatively more robust to residual pleiotropy than the effect of HDL on CAD, which could be overturned by residual pleiotropy as strong as LDL and triglycerides.

**Current proposals for MR "sensitivity analyses" can lead to false positive findings in the presence of small residual biases in large samples.** Prevailing proposals for sensitivity analyses of MR studies have focused on replacing traditional instrumental variable assumptions with alternative assumptions about how pleiotropy operates, such as the InSIDE[39,40] (Instrument Strength Independent of Direct Effect) or ZEMPA[41,42] (Zero Modal Pleiotropy) assumptions. Although an improvement of traditional MR, under the presence of residual population stratification, batch effects, and certain forms of pleiotropy, such approaches may still lead to statistically significant false findings given large enough samples. Therefore, the sensitivity statistics and exercises we propose here can be a useful complement to those alternative analyses.

To demonstrate this, we performed a simulation study in which the InSIDE assumption is only slightly violated through

small pleiotropic effects via confounders of the exposure and outcome trait. Our simulation largely follows the same specification of previous work[40,66,67], with the following data-generating process (DGP):

$$W_i = \sum_{j=1}^{J} \phi_j G_{ij} + \varepsilon_{i,W}, \qquad X_i = \sum_{j=1}^{J} \delta_j G_{ij} + \varepsilon_{i,X} \qquad (2)$$

$$D_i = \sum_{j=1}^{J} \beta_j G_{ij} + X_i + W_i + U_i + \varepsilon_{i,D} \qquad (3)$$

$$Y_i = \tau D_i + \eta X_i + \gamma W_i + U_i + \varepsilon_{i,Y} \qquad (4)$$

where $D_i$ is the exposure trait for individual $i$; $Y_i$ is the outcome trait; $W_i$ is an unobserved trait, and $X_i$ an observed trait, both carriers of pleiotropy in a way that violates the InSIDE assumption. The genetic variants $G_{ij}$ are drawn independently from a Binomial distribution, Binom(2, 1/3); the remaining error terms $U_i$, $\varepsilon_{i,W}$, $\varepsilon_{i,X}$, $\varepsilon_{i,D}$ and $\varepsilon_{i,Y}$ are drawn from standard gaussians. The DAG corresponding to the model of Eqs. (2)–(4) is shown in Fig. 4.

We set the number of variants $J = 90$, similar to our previous BMI analysis, and consider genetic effects drawn from an uniform

distribution from 0.01 to 0.05 for $\phi_j$, $\delta_j$ and $\beta_j$. The parameters $\eta$ and $\gamma$ give further control to the level of pleitropy, and here we set both to 0.05. To put this value in context, for the usual simulated sample size considered in previous work (10,000–30,000 individuals), this level of pleitropy is small enough that it does not meaningfully affect type I errors for MR-Egger. Here, however, we simulate larger sample sizes, similar to those found in large genetic databases, ranging from $N = 150{,}000$ to $N = 450{,}000$ individuals. Also note that we have a violation of the ZEMPA assumption[41,42], since the modal value of pleitropy, though very small, is not zero.

We investigated the performance of alternative MR methods in a two-sample Mendelian randomization setting, meaning that only summary level data was used in the analyses, and the genetic associations with the exposure trait and the outcome trait were obtained in separate simulated data (both with the same sample size $N$). Table 3 shows the results of 1000 simulations of the data-generating process for each of the sample sizes, considering two cases: (i) a true null causal effect with $\tau = 0$; (ii) and a true positive causal effect of $\tau = 0.1$. Note that $X_i$ and $W_i$ have similar strengths—a fact that, if known, can be exploited for sensitivity analysis.

We first focus on the case of a null causal effect. The first columns of the table shows the proportion of cases in which the null hypothesis of zero effect was rejected, using different MR methods: (i) the traditional inverse variance weighted (IVW); (ii) MR-PRESSO[40]; (iii) MR-Egger[39]; (iv) MR-GENIUS[43]; (iv) MR-MBE[41]; and (v) MR-Mix[42]. Since the true causal effect is zero, these results indicate the proportion of false positives. We see that IVW, MR-PRESSO and MR-GENIUS give similar results with a virtually 100% false positive rate for all sample sizes, and that MR-Egger, MR-MBE and MR-Mix start with false positive rates of 16%, 28% and 61% for $N = 150{,}000$, and this rate grows up to 46%, 80% and 69% at $N = 4500{,}000$, respectively. Note MR-GENIUS requires the assumption of a heteroscedastic first stage, which is not satisfied in the DGP (the performance of MR-GENIUS when gradually increasing the level of heteroscedasticity is assessed in the Supplementary Information).

Next, the last four columns show how the sensitivity exercises proposed in this paper could help interpreting the results in such cases. The "Critical $k$" columns show the 5th and 95th percentiles, across simulations, of the multiple of the strength of W relative to X that would be necessary to explain away the observed association. Visually, the critical $k$ is the value of the relative strength $k$ that would be necessary to bring the benchmark bound up to a critical line in the sensitivity contour plot (see Methods). For instance, in the previous DBP analysis, the critical $k$ for a confounder $k$ times as strong as observed principal components (Fig. 3a) is about 14, whereas for deprivation (Fig. 3b) this is about 0.5. Here, note that in 95% of the simulated scenarios the critical $k$ is safely below 1. Therefore, if the researcher suspects that residual pleitropy could be as strong as that of X, she would correctly be warned that such biases are strong enough to be problematic. The last two columns show the 5th and 95th percentiles, across simulations, of the robustness values for the association of the genetic instrument with the outcome. Note these always remain roughly below 0.6%, correctly warning the researcher that residual biases of those magnitudes are capable of overturning those MR findings.

We now turn to the second scenario, in which there is a true positive causal effect of D on Y. Here all MR methods correctly reject the null hypothesis of zero effect from 84% to 100% of the time. The challenge in this setting, thus, comes not from rejecting the null hypothesis, but from the fact that potential critics of the study could correctly be skeptical of the results, and conjecture that the reason why the null was rejected was simply due to residual pleitropic pathways. To mitigate those concerns, the researcher could again use the bounding procedure, and in over 95% of the simulations she would conclude that one would still reject the null, even when allowing for residual pleitropy as strong as that due to the observed X. Likewise, the results for the RV show that a researcher obtains a robustness value above 2.2% in at least 95% of the settings, meaning that, in all such cases, the critic would need to argue that biases of at least these magnitudes are plausible in order to forcefully dismiss the observed MR finding.

We can further illustrate the impact of small biases in large databases with the simulation of Table 4. Here we fix the sample size at 450,000 individuals and we also fix the true causal effect at $\tau = 0$. But now we concentrate pleitropic effects violating the InSIDE assumption on only a subset of the variants, allowing from 20 up to all of the 90 genetic variants to be fully valid instruments. Observe that false-positive rates remains excessively high, except when all variants are indeed valid. Of particular interest are the results of MR-MBE and MR-Mix, since, in theory, the ZEMPA assumption holds in all scenarios (i.e, the mode of pleitropic effects is zero in the population). In practice, however,

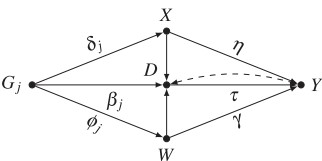

**Fig. 4 DAG of the data-generating process for the simulation study (Eqs. (2)–(4)).** For simplicity, only one $G_j$ is shown.

| Table 3 Simulation of weak pleiotropic pathways violating the InSIDE assumption. | | | | | | | | | | |
|---|---|---|---|---|---|---|---|---|---|---|
| | Proportion of rejections of the null ($\alpha = 5\%$) | | | | | | Critical $k$ | | $RV_{\alpha=0.05}$ | |
| Sample size | IVW | PRESSO | Egger | GENIUS | MBE | Mix | 5th | 95th | 5th | 95th |
| Scenario 1: true null causal effect ($\tau = 0$) | | | | | | | | | | |
| 150,000 | 100% | 100% | 16% | 100% | 28% | 61% | 0.00 | 0.85 | 0.0% | 0.6% |
| 300,000 | 100% | 100% | 35% | 100% | 60% | 68% | 0.02 | 0.87 | 0.0% | 0.6% |
| 450,000 | 100% | 100% | 46% | 100% | 80% | 69% | 0.17 | 0.88 | 0.0% | 0.6% |
| Scenario 2: true positive causal effect ($\tau = 0.1$) | | | | | | | | | | |
| 150,000 | 100% | 100% | 97% | 100% | 100% | 84% | 1.06 | 1.55 | 2.2% | 3.1% |
| 300,000 | 100% | 100% | 100% | 100% | 100% | 84% | 1.19 | 1.59 | 2.5% | 3.1% |
| 450,000 | 100% | 100% | 100% | 100% | 100% | 84% | 1.23 | 1.59 | 2.6% | 3.1% |

All methods are run with default parameter choices.
*RV* Robustness Value.

**Table 4 Simulation in which pleiotropic pathways are concentrated on only a subset of variants.**

| # of valid variants | Proportion of rejections of the null ($\alpha = 5\%$) | | | | | | Critical $k$ | | $RV_{\alpha=0.05}$ | |
|---|---|---|---|---|---|---|---|---|---|---|
| | IVW | PRESSO | Egger | GENIUS | MBE | Mix | 5th | 95th | 5th | 95th |
| 20 | 100% | 100% | 99% | 100% | 88% | 67% | 0 | 0.9 | 0% | 0.5% |
| 40 | 100% | 100% | 99% | 100% | 91% | 60% | 0 | 0.9 | 0% | 0.4% |
| 60 | 99% | 99% | 98% | 100% | 87% | 56% | 0 | 0.9 | 0% | 0.3% |
| 80 | 57% | 56% | 68% | 100% | 55% | 25% | 0 | 0.7 | 0% | 0.1% |
| 90 | 4.3% | 4.6% | 3.5% | 100% | 0.5% | 2.1% | 0 | 0 | 0% | 0 |

Sample size of 450,000 and $\tau = 0$. All methods are run with default parameter choices.
RV Robustness Value.

the mode is not exactly zero in finite samples, and certain default choices of tuning parameters for these procedures (such as the bandwith) may still lead to bias.

The phenomenon demonstrated in these simulations is simply the well known but often overlooked fact that, with large enough data, virtually any residual bias will eventually be statistically significant. It is for that reason that alternative analyses leveraging exact, sharp assumptions, such as the ones just described, are bound to lead to false positives with large enough genetic databases, unless their modified identification assumptions also hold exactly (or, of course, if they are extremely under-powered). In contrast, the sensitivity statistics we propose here, such as the partial $R^2$ and the RV, are directly quantifying the strength of biases needed to overturn a finding—and they will simply converge to their population values as the sample size increases. In the Supplementary Information we provide an additional set of simulations varying the relative strength of W to be 2, 3 or 4 times stronger than that of X, under the null hypothesis of zero causal effect. In that scenario, the sensitivity analysis now correctly reveals that it would take stronger confounding (about 2, 3 or 4 times as strong as X) to explain away the results.

## Discussion

We have described a suite of sensitivity analysis tools for performing valid MR inferences under the presence of residual biases of any postulated strength. The approach we proposed here starts from the premise that all MR studies will be imperfect in some way or another, but also that a study does not have to be perfect in order to be informative—what matters is not whether certain assumptions hold exactly, but the extent to which certain conclusions are robust to violations of those assumptions, and whether such strong violations are plausible.

We showed how two simple sensitivity statistics, the partial $R^2$ and the robustness value (RV), can be used to easily communicate the minimum strength of residual biases necessary to invalidate the results of a MR study. Since researchers are already well advised to report the partial $R^2$ of the genetic instrument with the exposure trait, routinely reporting the partial $R^2$ of the genetic instrument with the outcome trait and the robustness value is but a small addition to current practices, and can greatly improve the transparency regarding the robustness of MR findings. These sensitivity statistics have roots on a strong tradition in sensitivity analysis, dating back to at least Cornfield[68], that seeks to derive the minimal strength that unobserved variables must have in order to logically account for the observed association. The RV and the partial $R^2$, first developed in Cinelli and Hazlett[46], can in fact be interpreted as generalized "Cornfield conditions" for partial regression coefficients.

Related work by VanderWeele and Ding[69] introduced the E-value, establishing generalized Cornfield conditions for the risk-ratio. The E-value has also been proposed for sensitivity analysis of MR studies[70]. Cinelli and Hazlett[46,p.61] provide some

discussion of the differences between the RV and the E-value. Briefly, for effect measures such as $\beta_{YZ|XW}$ and $\beta_{YD|XW}$ (the targets of inference of the traditional IV estimand), the E-value provides an approximation, while the RV and the partial $R^2$ are exact. Moreover, while the RV parameterizes the association of the confounder in terms of residual variance explained, the E-value parameterizes those in terms of risk ratios. Whether one scale is preferable over the other is context dependent. In particular, we note that in MR and genetics, effect measures in terms of partial $R^2$ seem to be ubiquitous (e.g. see notions such as "heritability"). In the Supplementary Information we provide the E-values for our running example and further discussion.

We also showed that, whenever researchers are able to argue that, although not perfect, they have credibly accounted for most of the population structure with genomic principal components, most of possible batch effects with technical covariates, and have measured known important pleiotropic pathways, this knowledge can be leveraged to formally bound the worst possible inferences due to residual biases. Such bounding exercises can be an important piece of the scientific debate when arguing in favor or against the robustness of a certain finding. A seemingly related tool to the contour plots with benchmark bounds we discussed in this paper is known in epidemiology as "bias-components" plot[71,72]. Although such plots can be useful for understanding and decomposing the difference between an IV estimate including and excluding observed variables, and contrasting this to "usual" estimates (adjusting for observed confounders), again including and excluding observed variables, these plots do not provide formal sensitivity analysis due to unmeasured confounding. If used for that purpose, they can lead users to erroneous conclusions even when unobserved variables are assumed to be identical to observed ones (see example in the Supplementary Information).

In this paper we focused on biases due to residual population structure, batch effects or horizontal pleiotropy. Nevertheless, so long as the identifying functional of the MR study is the traditional IV estimand, as given by Eq. (1), the sensitivity analysis we propose here is still applicable, as it is agnostic to the particular structure that creates bias, or the particular causal interpretation of the target of inference. This encompasses biases due to several different DAG structures, such as time-varying treatments and outcomes, linkage disequilibrium[21,69], and even selection bias[70] (see examples in the Supplementary Information).

Finally, we remind readers that sensitivity analysis tools, such as the ones we propose here, are not aiming to replace contemporary MR methods (such as MR-PRESSO[40] MR-Egger[39], MR-GENIUS[43], MR-MBE[41], MR-Mix[42] and others), nor aiming to be a substitute to expert judgment. On the contrary, these tools can be used as a useful complement to traditional MR analyses, by aiding experts in leveraging certain types of knowledge that would have been otherwise neglected, such as judgments regarding the maximum plausible strength of residual biases, or

knowledge regarding the relative importance of certain causal pathways. In sum, strong conclusions from Mendelian randomization studies still need to rely on the quality of the research design, substantive understanding both of the genetic variants as well as the traits under investigation, and the triangulation of evidence from multiple sources and methods. Extension of these sensitivity analysis tools to the context of generalized linear models, or to explicitly leverage the information created by multiple instruments, is currently under work.

## Methods

### Study design and participants

*Study population.* The UK Biobank[73] is a resource that links genetic data to a variety of physiological and social traits in a cohort of 503,325 British people aged 37–73 years. It has been a valuable resource for estimating causal effects of exposures on a multitude of outcomes using MR[56–58]. We filtered the data to only include people with self-reported white British ancestry who were not closely related, (e.g. no first, second, or third degree relatives), as defined by pairs of individuals who had a kinship coefficient < $(1/2)^{(9/2)}$ (following[74]), leaving 291,274 people. We also removed individuals who were not measured for BMI (non-impedence). For our analysis of the Lyall et al.[57] study, we also excluded patients who responded to a question on whether they were taking anti-hypertensive medication with "don't know."

*Polygenic risk score.* The Polygenic Risk Scores (PRS) was constructed in the same manner as in Lyall et al.[57]. This PRS score was derived from 97 SNPs that were genome-wide significantly associated with BMI in the GIANT consortium study[59]. Two of these SNPs were not directly genotyped in the UK Biobank, and two failed Hardy-Weinberg equilibrium, leaving 93 SNPs to comprise the PRS. The PRS was computed as a weighted score based on these SNPs, with the weights derived from the effect estimated reported by GIANT ($\beta$ per 1-SD unit of BMI)[57,59]. We used the exact same weights computed by Lyall et al.[57] (Supplementary Data 1).

*Exposure, outcome and control traits.* During the initial visit to the UK Biobank assessment center, height was measured to the nearest centimeter using a Seca 202 device and weight was measured to the nearest 0.1 kg using a Tanita BC418MA body composition analyzer. These measurements were subsequently used to calculate body mass index (BMI), in kg/m² (field category ID: 21001). The two outcomes of interest were the Townsend deprivation index and diastolic blood pressure. The Townsend deprivation index was calculated using the postcode of the participant at the time of recruitment (field category ID: 189). Diastolic blood pressure was obtained by an automated reading from an Omron blood pressure monitor (field category ID: 4079). In our analyses, we adjusted for age, sex, assessment center, genetic batch effects, drinking and smoking status, given by the following variables: "Sex" (field category ID: 31); "Age when attended assessment center" (field category ID: 21003); "UK Biobank assessment center" (field category ID: 54); "Genotype measurement batch" (field category ID: 22000); "Smoking status" (field category ID: 20116); "Frequency of drinking alcohol" (field category ID: 20414); "Alcohol intake frequency" (field category ID: 1558).

### Statistical methods

*Traditional Mendelian Randomization.* Suppose we are interested in assessing the causal effect of an exposure trait $D$ on an outcome trait $Y$, by performing a Mendelian Randomization study with a polygenic risk score (PRS) $Z = \sum_{j=1}^{k} \beta_j G_j$ (comprised of a linear combination of SNPs $G_j$ with weights $\beta_j$) as the putative instrumental variable. Note the weights $\beta_j$ of the PRS could have been obtained either from external data (such as a previous GWAS), or via cross-validation as well as other methods[9]. To give credibility to the study, the researcher considers a set of observed control covariates $X$ that accounts for potential MR violations of population stratification, batch effects and horizontal pleiotropy[20]. That is, $X$ consists of,

$$X = \{X_{ps}, X_{batch}, X_{hp}, X_{ind}\}$$

Where $X_{ps}$ denotes the variables to adjust for population stratification, such as, for instance, genomic principal components; $X_{batch}$ denotes variables to adjust for batch effects, for example, indicator variables for the assessment center and genotype batches; $X_{hp}$ denotes measured variables which are suspected to be capable of blocking suspected pleiotropic pathways; and, finally, $X_{ind}$ are participant characteristics that are usually included in MR, such as the age and sex of the individual.

The traditional MR estimate of the causal effect of $D$ on $Y$, here denoted by $\hat{\tau}_{res}$, would consist of the ratio of the genetic association with the outcome trait, $\hat{\beta}_{YZ|X}$, and the genetic association with the exposure trait, $\hat{\beta}_{DZ|X}$, after adjusting for observed covariates $X$, namely,

$$\hat{\tau}_{res} = \frac{\hat{\beta}_{YZ|X}}{\hat{\beta}_{DZ|X}}$$

Confidence intervals that have nominal coverage regardless of instrument strength can be obtained via Fieller's theorem[53] or via the Anderson–Rubin regression[52]. These confidence intervals can be of three forms: (i) a connected closed interval; (ii) the union of disjoint unbounded intervals; or, (iii) the whole real line.

*Violation of traditional assumptions.* The traditional MR estimate, $\hat{\tau}_{res}$, however adjusts for $X$ only, and it is unlikely that $X$ controls for all possible threats to the study validity. Instead, the researcher would have preferred to have also adjusted for additional unobserved variables $W$ to satisfy the MR assumptions. For instance, we would like to have controlled for the true population indicators $W_{ps}$ instead of its approximation as recovered by the principal components $X_{ps}$; likewise, the researcher suspects that $X_{hp}$ is not enough to block all pleiotropic pathways, and would have liked to have further adjusted for covariates $W_{hp}$.

In sum, instead, of performing the MR analysis using $X$ alone, resulting in $\hat{\tau}_{res}$ as our MR estimate, the researcher would have wanted to compute instead

$$\hat{\tau} = \frac{\hat{\beta}_{YZ|XW}}{\hat{\beta}_{DZ|XW}}$$

which adjusts for the extended set of covariates $\{X, W\}$, such that $Z$ is a valid instrument for estimating a specific target causal effect of $D$ on $Y$, conditional on $\{X, W\}$. Likewise, confidence intervals should have also been computed adjusting for $\{X, W\}$. How would accounting for the omitted variables $W$ have changed our inferences regarding the causal effect of $D$ on $Y$?

*The sensitivity analysis of the MR estimate can be reduced to the sensitivity of the genetic associations.* We now explain how to perform sensitivity analysis within the Anderson–Rubin (AR) approach[52], which as we show is also numerically equivalent to Fieller's proposal[53] when considering a single instrumental variable $Z$. Here we take an exact algebraic approach—that is, all results here hold both for sample or population estimates.

Let $Y$ and $D$ denote $(n \times 1)$ vectors containing the outcome and exposure of interest for each of the $n$ observations, respectively. Now let $\tau$ denote the causal effect of interest, and define a new variable $Y_{\tau_0} := Y - \tau_0 D$, in which we subtract from $Y$ the causal effect of $D$, considering a hypothetical value for $\tau$, say, $\tau_0$. Next consider the following linear regression,

$$Y_{\tau_0} = \hat{\phi}_{\tau_0} Z + X\hat{\eta}_{\tau_0} + W\hat{\gamma}_{\tau_0} + \hat{\varepsilon}_{\tau_0} \tag{5}$$

Where $Z$ is a $(n \times 1)$ vector with the genetic instrument; $X$ is a $(n \times p)$ matrix of observed covariates, including the constant; and $W$ is a $(n \times k)$ matrix of unobserved covariates the analyst wished to have measured in order to make $Z$ a valid instrument. Here $\hat{\phi}_{\tau_0}$, $\hat{\eta}_{\tau_0}$, $\hat{\gamma}_{\tau_0}$ are the OLS coefficient estimates of the regression of $Y_{\tau_0}$ on $Z, X, W$, and $\hat{\varepsilon}_{\tau_0}$ its corresponding residual.

Note that, if $\tau = \tau_0$ and if $Z$ is valid instrument conditional on $X, W$, then we must have that $Y_{\tau_0} \perp\!\!\!\perp Z | X, W$, and thus that $\phi_{\tau_0} = 0$. Following this logic, the AR confidence interval with confidence level $1 - \alpha$ is thus defined as all values of $\tau_0$ such that we cannot reject the null hypothesis $H_0 : \phi_{\tau_0} = 0$ at the chosen significance level. More precisely,

$$CI_{1-\alpha}(\tau) = \left\{ \tau_0 ; t_{\phi_{\tau_0}}^2 \leq t_{\alpha, df}^{*2} \right\} \tag{6}$$

Where $t_{\phi_{\tau_0}}$ is the t-value for testing the null hypothesis that $H_0 : \phi_{\tau_0} = 0$ and $t_{\alpha, df}^*$ is the critical threshold of the t distribution for a significance level $\alpha$ and df degrees of freedom. This confidence interval can be obtained analytically as a function of the genetic association with the exposure and the genetic association with the outcome, which is now useful to write out explicitly.

By appealing to the Frisch–Waugh–Lovell (FWL) theorem[75–77], we can write $\hat{\phi}_{\tau_0}$ as,

$$\hat{\phi}_{\tau_0} = \frac{\text{cov}\left(Y^{\perp XW} - \tau_0 D^{\perp XW}, Z^{\perp XW}\right)}{\text{var}\left(Z^{\perp XW}\right)}$$
$$= \frac{\text{cov}\left(Y^{\perp XW}, Z^{\perp XW}\right)}{\text{var}\left(Z^{\perp XW}\right)} - \tau_0 \frac{\text{cov}\left(D^{\perp XW}, Z^{\perp XW}\right)}{\text{var}\left(Z^{\perp XW}\right)} \tag{7}$$
$$= \hat{\beta}_{YZ|XW} - \tau_0 \hat{\beta}_{DZ|XW}$$

Where $Y^{\perp XW}$ denotes the variable $Y$ after removing the components linearly explained by $X$ and $W$, and $\hat{\beta}_{YZ|XW}$ denotes the regression coefficient of $Z$ on $Y$ (the genetic association with the outcome) after adjusting for both $X$ and $W$; $\hat{\beta}_{DZ|XW}$ denotes the regression coefficient of $Z$ on $D$ (the genetic association with the exposure) after adjusting for $X$ and $W$. Likewise, the estimated variance of $\hat{\phi}_{\tau_0}$ can

be written as,

$$
\begin{aligned}
\widehat{\mathrm{var}}(\hat{\phi}_{\tau_0}) &= \frac{\mathrm{var}\left(Y^{\perp ZXW} - \tau_0 D^{\perp ZXW}\right)}{\mathrm{var}\left(Z^{\perp XW}\right)} \times \mathrm{df}^{-1} \\
&= \widehat{\mathrm{var}}\left(\hat{\beta}_{YZ|XW}\right) + \tau_0^2\,\widehat{\mathrm{var}}\left(\hat{\beta}_{DZ|XW}\right) \\
&\quad - 2\tau_0\,\widehat{\mathrm{cov}}\left(\hat{\beta}_{YZ|XW}, \hat{\beta}_{DZ|XW}\right)
\end{aligned}
\tag{8}
$$

To construct the confidence interval of Eq. (6), we thus need to find all values of $\tau_0$ such that the following inequality holds,

$$
\frac{\hat{\phi}_{\tau_0}^2}{\widehat{\mathrm{var}}\left(\hat{\phi}_{\tau_0}\right)} \le t_{\alpha,\mathrm{df}}^{*2} \;\Rightarrow\; \hat{\phi}_{\tau_0}^2 - \widehat{\mathrm{var}}\left(\hat{\phi}_{\tau_0}\right)t_{\alpha,\mathrm{df}}^{*2} \le 0
\tag{9}
$$

Squaring and rearranging terms we obtain the following quadratic inequality,

$$
\underbrace{\left(\hat{\beta}_{DZ|XW}^2 - \widehat{\mathrm{var}}\left(\hat{\beta}_{DZ|XW}\right) \times t_{\alpha,\mathrm{df}}^{*2}\right)}_{a}\tau_0^2
$$
$$
+ \underbrace{2\left(\widehat{\mathrm{cov}}\left(\hat{\beta}_{YZ|XW}, \hat{\beta}_{DZ|XW}\right) \times t_{\alpha,\mathrm{df}}^{*2} - \hat{\beta}_{YZ|XW}\hat{\beta}_{DZ|XW}\right)}_{b}\tau_0
$$
$$
+ \underbrace{\left(\hat{\beta}_{YZ|XW}^2 - \widehat{\mathrm{var}}\left(\hat{\beta}_{YZ|XW}\right) \times t_{\alpha,\mathrm{df}}^{*2}\right)}_{c} \le 0
\tag{10}
$$

These conditions are exactly Fieller's solution[53] to the confidence interval of the ratio $\tau = \frac{\beta_{YZ|XW}}{\beta_{DZ|XW}}$.

Our task has thus simplified to find all values of $\tau_0$ that makes the above quadratic equation, with coefficients $a$, $b$ and $c$, non-positive. But here we have special interest in two specific cases: (i) when the confidence interval for $\tau$ is unbounded; and, (ii) when the confidence interval for $\tau$ includes zero.

Let us first consider the case of unbounded confidence intervals. Note this happens when $a < 0$, which means the quadratic curve in Eq. (10) will be concave (will have a "$\cap$" shape)—as we increase $\tau_0$ to plus or minus infinity, the inequality is bound to hold and the confidence interval will be unbounded. Also note that $a < 0$ if, and only if,

$$
\hat{\beta}_{DZ|XW}^2 - \widehat{\mathrm{var}}\left(\hat{\beta}_{DZ|XW}\right) \times t_{\alpha,\mathrm{df}}^{*2} \le 0 \;\Rightarrow
$$
$$
\frac{|\hat{\beta}_{DZ|XW}|}{\widehat{\mathrm{se}}\left(\hat{\beta}_{DZ|XW}\right)} = |t_{\hat{\beta}_{DZ|XW}}| \le t_{\alpha,\mathrm{df}}^{*}
\tag{11}
$$

That is, the confidence interval for $\tau$ will be unbounded if and only if we cannot reject that the genetic association with the exposure is zero.

We now turn our attention to the null hypothesis of zero effect, that is, $H_0: \tau = 0$. Notice in this case the first two terms of the quadratic equation, $a$ and $b$, vanish. What we have left is only the term $c$ which will be negative if, and only if,

$$
\hat{\beta}_{YZ|XW}^2 - \widehat{\mathrm{var}}\left(\hat{\beta}_{YZ|XW}\right) \times t_{\alpha,\mathrm{df}}^{*2} \le 0 \;\Rightarrow
$$
$$
\frac{|\hat{\beta}_{YZ|XW}|}{\widehat{\mathrm{se}}\left(\hat{\beta}_{YZ|XW}\right)} = |t_{\hat{\beta}_{YZ|XW}}| \le t_{\alpha,\mathrm{df}}^{*}
\tag{12}
$$

In other words, the null hypothesis of zero effect for the causal effect is not rejected if, and only if, the null hypothesis of zero association between the instrument $Z$ with the outcome $Y$ is also not rejected.

We have thus simplified the sensitivity analysis of the MR estimate to the sensitivity analysis of the two genetic associations. If $W$ is strong enough to explain away the genetic association with the exposure, then $W$ is strong enough to make the causal effect arbitrarily large in either direction. If $W$ is strong enough to explain away the genetic association with the outcome trait, than $W$ is strong enough to explain away the MR estimate. This is summarized in Table 5.

Since we have reduced the problem of sensitivity analysis of MR to the problem of sensitivity analysis of the genetic associations, we can leverage all tools of Cinelli and Hazlett[46] for our problem. To conclude we thus review the main sensitivity analysis results of Cinelli and Hazlett, in the context of the genetic association with the outcome. All results below, of course, also apply to the genetic association with the exposure, by just replacing $Y$ with $D$ where appropriate.

*Sensitivity formulas for the genetic associations.* Consider first a univariate $W$ and let $R_{ZW|X}^2$ denote the partial $R^2$ of $W$ with the genetic instrument and let $R_{YW|ZX}^2$ denote the partial $R^2$ of $W$ with the outcome trait. Given the observed genetic association $\hat{\beta}_{YZ|X}$ and its estimated standard error $\widehat{\mathrm{se}}(\hat{\beta}_{YZ|X})$, adjusting for $X$ alone, the estimate and standard error we would have obtained further adjusting for W can be recovered with[46],

$$
\hat{\beta}_{YZ|XW} = \hat{\beta}_{YZ|X} \pm \widehat{\mathrm{se}}\left(\hat{\beta}_{YZ|X}\right)\sqrt{\frac{R_{YW|ZX}^2 R_{ZW|X}^2}{1 - R_{ZW|X}^2}(\,\mathrm{df}\,)}
\tag{13}
$$

and,

$$
\widehat{\mathrm{se}}\left(\hat{\beta}_{YZ|XW}\right) = \widehat{\mathrm{se}}\left(\hat{\beta}_{YZ|X}\right)\sqrt{\frac{1 - R_{YW|ZX}^2}{1 - R_{ZW|X}^2}\left(\frac{\mathrm{df}}{\mathrm{df} - 1}\right)}
\tag{14}
$$

Where here now df denote the degrees of freedom of the AR regression actually run. These formulas allow us to investigate how the estimate, standard error, $t$-values, $p$-values or confidence intervals would have changed, under a confounder W of any postulated strength, as parameterized by $R_{ZW|X}^2$ and $R_{YW|ZX}^2$. For a singleton W these formulas are exact, and for multivariate $W$, it can further be shown that these formulas are conservative, barring an adjustment on the degrees of freedom[46] (that is, these are the worse biases a multivariate $W$ could cause). These formulas form the basis of the contour plots shown in Fig. 2.

*Bounds on the partial $R^2$ of W based on observed covariates.* Where investigators are unable to make direct claims on the strength of $W$, it may be helpful to consider relative claims, by positing, for instance, that $W$ is no stronger than some observed covariate $X_j$. For that, consider a confounder orthogonal to the observed covariates, ie., $W \perp X$ and define

$$
k_Z := \frac{R_{ZW|\mathbf{X}_{-j}}^2}{R_{ZX_j|\mathbf{X}_{-j}}^2}, \qquad k_Y := \frac{R_{YW|\mathbf{X}_{-j}Z}^2}{R_{YX_j|\mathbf{X}_{-j}Z}^2}.
\tag{15}
$$

where $\mathbf{X}_{-j}$ represents the vector of covariates $\mathbf{X}$ excluding $X_j$. Then the strength of $W$ can be bounded by[46],

$$
R_{ZW|\mathbf{X}}^2 = k_Z f_{ZX_j|\mathbf{X}_{-j}}^2, \quad R_{YW|ZX}^2 \le \eta^2 f_{YX_j|\mathbf{X}_{-j}Z}^2
\tag{16}
$$

where $\eta$ is a scalar computed from $k_Y$, $k_Z$ and $R_{ZX_j|\mathbf{X}_{-j}}^2$ (see Cinelli and Hazlett[46] for details) and $f^2 = \frac{R^2}{1-R^2}$ is the partial Cohen's $f^2$ statistic.

*Sensitivity statistics for routine reporting.* The previous results allow us to perform sensitivity analysis to confounding of any postulated strength. However, widespread adoption of sensitivity analysis benefits from simple metrics that users can report to quickly summarize the robustness of their results. With that in mind, Cinelli and Hazlett[46] introduced two sensitivity statistics for routine reporting: the Robustness Value (RV) and the partial $R^2$.

Let $f := |f_{Y\sim Z|\mathbf{X}}|$ denote the absolute value of the partial Cohen's $f$ of the genetic instrument with the outcome. Now also re-scale the critical threshold, $f_\alpha^* := |t_{\alpha,\mathrm{df}-1}^*|/\sqrt{\mathrm{df}-1}$, and define $f_\alpha := f - f_\alpha^*$. The robustness value $\mathrm{RV}_\alpha$ is defined as the minimal strength of association that $W$ must have, both with the genetic instrument $Z$ and the outcome trait $Y$, in order to make the genetic

**Table 5 The sensitivity of the Mendelian Randomization (MR) causal effect estimate can be decomposed into the sensitivity of its two components: the sensitivity of the genetic association with the exposure and the sensitivity of the genetic association with the outcome.**

| Sensitivity analysis | Interpretation |
|---|---|
| Of the genetic association with the exposure | The sensitivity of the genetic association with the exposure reveals the stability of the MR causal effect estimate. Biases strong enough to result in a failure of rejection that the genetic association with the exposure is zero, also lead to unbounded confidence intervals for the MR causal effect estimate. |
| Of the genetic association with the outcome | The sensitivity of the genetic association with the outcome is equivalent to the sensitivity of the MR causal effect estimate with respect to the zero null hypothesis. Biases strong enough to result in a failure of rejection that the genetic association with the outcome is zero equally result in a failure to reject the null hypothesis that the MR causal effect estimate is zero. |

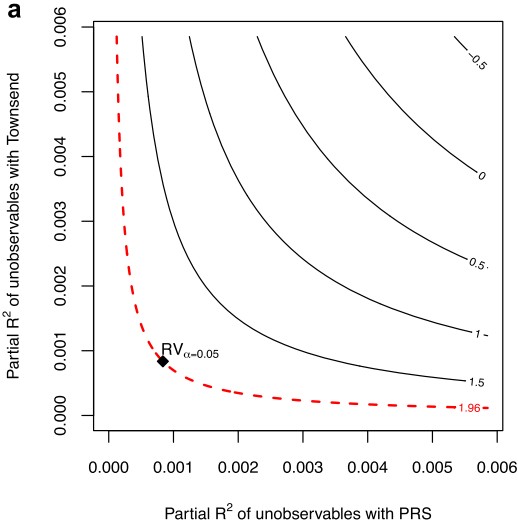
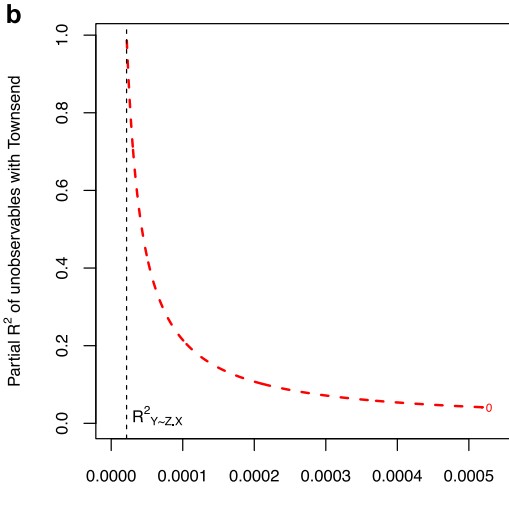

**Fig. 5 Visual depiction of the sensitivity statistics $RV_\alpha$ and $R^2_{YZ|X}$ in the sensitivity contours for Townsend deprivation index. a** Sensitivity contour showing the robustness value at the 5% significance level, $RV_{\alpha=0.05}$. **b** Sensitivity contour showing the partial $R^2$ of the genetic instrument with the outcome, $R^2_{YZ|X}$.

association with the outcome statistically insignificant. This is given by[46,47]

$$RV_\alpha = \begin{cases} 0, & \text{if } f_\alpha < 0 \\ \frac{1}{2}\left(\sqrt{f_\alpha^4 + 4f_\alpha^2} - f_\alpha^2\right), & \text{if } f_\alpha^* \leq f < f_\alpha^{*-1} \\ \frac{f^2 - f_\alpha^{*2}}{1+f^2}, & \text{otherwise}. \end{cases} \quad (17)$$

Any $W$ with both strength of associations below $RV_\alpha$ is not sufficiently strong to make the genetic association with the outcome statistically insignificant, and, thus, also not sufficiently strong to make the MR causal effect estimate statistically insignificant. On the other hand, any $W$ with both strength of associations above $RV_\alpha$ is sufficiently strong to do so.

Moving to the partial $R^2$, in addition to quantifying how much variation of the outcome trait is explained by the genetic instrument, the partial $R^2$ also tells us how robust the genetic association with the outcome is to an "extreme sensitivity scenario." More precisely, suppose that the unobserved variable $W$ explained all residual variance of the outcome trait. Then, for $W$ to bring the genetic association to zero, it must explain at least as much residual variation of the genetic instrument as the residual variation of the outcome trait that the genetic instrument currently explains[46]. Mathematically, if $R_{Y \sim W|ZX} = 1$, then for $W$ to make $\hat{\beta}_{YZ|XW} = 0$, we need to have that $R^2_{ZW|X} \geq R^2_{YZ|X}$.

It may be helpful to visualize both sensitivity statistics, $RV_\alpha$ and $R^2_{YZ|X}$, in a sensitivity contour plot. These are shown in Fig. 5, using as example the contours for testing the null hypothesis of zero effect of BMI on Townsend deprivation index. Starting with Fig. 5a, note that $RV_\alpha$ is the point of equal association of both sensitivity parameters that lies in the critical contour for statistical significance. The $RV_\alpha$ is thus a convenience reference point summarizing a contour of interest, quickly communicating the types of confounders that can and cannot be problematic. Now moving to Fig. 5b, our focus in the on the critical contour of zero (i.e., completely eliminating the point estimate). Note that the axes are now in a different scale: we "zoom in" on a neighborhood of the partial $R^2$ on the horizontal axis, so as to be able to show more smoothly the critical contour of zero until it reaches its maximum in the vertical axis. The partial $R^2$ then corresponds to the vertical line tangent to the critical contour of zero, which is never crossed. It thus summarizes the bare minimum strength of association that confounders need to have with the genetic instrument to fully account for the genetic association, regardless of how strongly such confounders are associated with the outcome trait.

The vertical line tangent to any contour for a given significance level $\alpha$ is given by the extreme robustness value, $XRV_\alpha$, derived in Cinelli and Hazlett[47], which is given by:

$$XRV_\alpha = \begin{cases} 0, & \text{if } f_\alpha \leq 0 \\ \frac{f^2 - f_\alpha^{*2}}{1+f^2}, & \text{otherwise}. \end{cases} \quad (18)$$

where $f^2 = \frac{R^2}{1-R^2}$ is Cohen's partial $f^2$ statistic. We thus have that the RV is the point of equal association of the contour, and the XRV the line tangent to that contour. Note when we set $\alpha = 1$ (that is, fully eliminating the point estimate, which corresponds to the zero contour) the XRV reduces to the partial $R^2$, i.e, $XRV_{\alpha=1} = \frac{f^2}{1+f^2} = R^2_{YZ|X}$.

For practical purposes in MR studies, we give preference to the simpler partial $R^2$. This is because the partial $R^2$ is a well known metric among MR researchers,

and it is also already common practice to report the partial $R^2$ of genetic instrument with the exposure (to assess the problem of "weak" instruments). Therefore, instead of introducing yet another metric, we simply suggest additionally computing the partial $R^2$ for the genetic association with the outcome, which comes naturally in this setting. Finally, we also note that all sensitivity statistics are simply transformations of the partial $R^2$ (through Cohen's $f$).

*Critical k.* Consider an unobserved variable $W$ with $k_Z = k_Y = k$ (of Eq. (15)). The "critical $k$" is the value of $k$ such that it brings the adjusted $t$-value for testing the null hypothesis of zero effect to the critical threshold of interest (say $\approx 1.96$ for $\alpha = 0.05$). Although we did not find a simple closed-form solution for the critical $k$ (such as the one we have for the robustness value), it can be easily found numerically.

**Reporting summary.** Further information on research design is available in the Nature Research Reporting Summary linked to this article.

## Data availability

The UK Biobank data are available under restricted access due to privacy laws. Access can be obtained by application at: http://www.ukbiobank.ac.uk/. Data from the UK Biobank was accessed as part of application 33127.

## Code availability

The software MR-SENSEMAKR for R and replication code can be found at: https://github.com/carloscinelli/mrsensemakr

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

## Acknowledgements

N.L. would like to acknowledge the support of National Science Foundation grant DGE-1829071 and National Institute of Health grant T32 EB016640. S.S. was supported in part by NSF grants III-1705121, III-2106908, CAREER-1943497, and NIH R35GM12505.

## Author contributions

C.C. developed the method, performed the simulation study, and designed the figures. N.L. and B.L.H. performed the UK Biobank data cleaning and analysis. C.C., N.L., and B.L.H. collaborated on the software development. C.C., N.L., B.L.H., S.S., and E.E. collaborated on the design of the experiments, interpretation of results, and writing of the paper. S.S. and E.E. obtained funding for the study. S.S. obtained access to the UK Biobank data.

## Competing interests

The authors declare no competing interests.
