## [Peer Review File · Nature Communications]

Title: Robust Mendelian randomization in the presence of residual population stratification, batch effects and horizontal pleiotropyREVIEWER COMMENTS

Reviewer #1 (Remarks to the Author):

Summary:

The authors propose an approach for quantifying the robustness of a Mendelian randomization analysis based on a polygenic risk score. This is an important issue, and the proposed method is elegant and produces convenient visualizations. I have some comments (below) but overall think this is really nice work.

Major Comments:

1. Many confounders may explain a much larger proportion of variance than any measured confounder including PCs. This can happen for example when two traits have a high genetic correlation or when the PRS used as the instrument is constructed from a small number of variants, some of which have large pleiotropic effects. The authors do state that “Bench- marks for horizontal pleiotropy, on the other hand, require specific knowledge of the aetiology of the disease under study, or of the social process under investigation.” However, I think this issue is worth a little more discussion. In particular, I think that PCs are probably bad benchmarks for pleiotropic effects (though they are fine for population structure effects).

2. Continuing the thought of point 1, this could be explored further in the real data example by looking at “messier” examples. One trait pair to consider would be HDL cholesterol and coronary artery disease. This is often considered an MR false positive (though there is some argument) that arises due to pleiotropy between HDL and other CAD risk factors including LDL, triglycerides, and BMI. I expect the pleiotropic confounding in this case would be much stronger than in the BMI -> townsend index case so it would be an interesting comparison.

3. This method applies only to MR using a single variant (probably a polygenic score), making it impossible to consider effect. Most methods that relax MR assumptions including MR-PRESSO, GSMR (Zhu et al, 2018), CAUSE (Morrison et al, 2020), and MR-MIX (Qi and Chatterjee, 2019) do so by assuming that a pleiotropic confounder will induce heterogeneity in the $\beta_{\text{hat_outcome}_j} / \beta_{\text{hat_exposure}_j}$ where j indexes the variant. This heterogeneity is then an independent piece of information to the data used in the proposed robustness metrics. I think it would be good to include a more nuanced discussion of this issue. Additionally, since robustness metrics cannot protect against pleiotropy that is stronger than existing benchmarks, would it be reasonable for investigators to deploy a hybrid approach looking at results from both types of methods?

4. Following from above, in the simulations, pleiotropic effects are spread evenly over all 90 variants, meaning that there is no or little heterogeneity in the causal effects implied by each individual variant. In this scenario, MR- PRESSO should perform nearly identically to plain IVW so, as presented, those results are really “strawman” results.

It would be nice to see additional simulations that a) concentrate pleiotropic effects on only a subset of variants and b) have a pleiotropic confounder that has stronger effects than the measured confounder. I appreciate that the authors point out that “The phenomenon demonstrated in the simulation is simply the well-known but often overlooked fact that, with large enough data, any residual bias will eventually be statistically significant.”

5. It seems plausible that smoking and alcohol consumption are colliders in the BMI-> Townsend index analysis, i.e. it seems plausible that higher Townsend deprivation index causes increased drinking or smoking and that there are genetic variants in the PRS linked to drinking and smoking. This would induce bias in the estimate obtained.

6. I think it would be useful to explain the relationship between the partial R^2 and RV values and the contour plots, since the verbal description is somewhat hard to follow. Is it correct that the dotted red line should have a vertical asymptote at the partial R^2 for the PRS-Outcome relationship and that the point (RV_alpha, RV_alpha) should lie on the red line? (this doesn't look correct in the BMI-Townsend index case)

7. It could be useful to also show the contour plot for partial R^2 PRS vs partial R^2 exposure for at least one of the BMI applications in order to illustrate the meaning of the summary stats for the PRS-Exposure relationship.

Minor Comments:

1. The last line of eq (7) follows immediately from the last line of eq. (6) so the preceding derivation could be omitted.

2. f_Z and f_Y appear in eq (15) but are not defined until the following paragraph. Should f_Z in the first part of eq (15) be R_Z ?

Reviewer #2 (Remarks to the Author):

The authors present methods and opportunities for bias and sensitivity analyses in MR. I greatly value the endeavor the authors. However, I have two major themes of concerns about the current version. One, the authors have not engaged with the work on similar methods already appearing in epidemiology, and thus it is hard to gauge the novelty of the contributions here (see some questions below). Two, I think the framing of the paper should better acknowledge the scope of the settings and types of bias being studied, and I'm concerned that when this scope gets clarified then these tools may not actually be as helpful as presented in real applications with all the complexities of human health. My comments below reflect these two themes.

1. How does your 'robustness value' and worst-case-bounds correspond to prior works on the E-value for MR (Swanson Vanderweele Epid.), bias component plots (Jackson Swanson Epid; Davies Epid.), and other approaches applied or described previously (e.g., Glymour et al. AJE 2012; Vanderweele et al. Epid.; the chapter on IVs in Modern Epidemiology's latest edition)? You may also want to compare your useful Figures with those that are common in pharmacoepidemiology settings, not for IVs necessarily but for quantifying confounding bias; and the original and canonical epi papers on this kind of topic from 70+ years ago by Cornfield, Bross, etc. (again, not in the setting of IVs).

2. You state boldly (as in you write it in bold) that current proposals fail whereas there toolbox does not. This again needs to acknowledge better the scope of the other proposals you are comparing to. It doesn't seem to reference the methods in #1 above, nor does it reference some of the novel 'robust' approaches like MR-GENIUS (Tchetgen Tchetgen et al.). It seems like you are primarily concerned in comparing to the popular MR-Egger. Though this limited comparison is illuminating given MR-Egger's popularity, it is not really engaging with the rich methods literature here and thus not advancing our collective understanding of how to do MR well with all tools available.

3. Can you comment on whether conclusions drawn from these approaches are always consistent with the observable data? Would applied researchers need to also consider the IV inequalities or other approaches (e.g., Balke and Pearl; Bonet; Diemer et al.), either prior to implementing all analyses or even integrating them somehow into your approaches?

4. Can you comment on how much each presented approach relies on assumptions of homogeneity, and if so what specific assumptions? Are these reasonable assumptions to be making in real MR studies? If yes, why? If no, how does that affect interpretability of these checks?

5. It needs to be clearer to the end user what specific DAGs might be valuable to consider here, as the same IV assumptions can be violated due to many different structures of violations (see Vanderweele et al. Epid. for some examples of the exclusion restriction; see Swanson Epid. for some examples of collider biases; just for a few examples). If these sensitivity analyses 'work' for thinking about multiple these types of biases, that should be clear; if they do not, the scope should be clear. Along the same lines, does this all work regardless of if Z, D, Y, X are continuous or dichotomous and whether Z is the risk score as described here vs. a vector of separate proposed instruments? This all is also important for contextualizing your findings against existing methods.

6. MR inherently is studying time-varying exposures with proposed IVs fixed at conception, yet 'the' DAG drawn by everybody is of a time-fixed or point exposure. How do your methods address this? What effect are you even assuming to be estimating - e.g., a lifetime effect (Labrecque AJE) or something else? Or simply a test of a sharp causal null hypothesis (Swanson et al. EJE)? I think the answer is just the test, given what you write near the beginning of page 3, but even so it is not clear if your approach contextualizes only a test for a sharp null of a point exposure and not a joint sharp null. In short, it seems like trying to understand bias means you first need to make the causal question clear, and that may mean acknowledging that this canonical IV DAG is not the best starting point for MR. Let me just

acknowledge in raising this specific point, however, that nearly all MR methods and applications are fuzzy about this, so this is not unique to the current paper.

Response to Reviewers—Nature Communications

Robust Mendelian randomization in the presence of residual population stratification, batch effects and horizontal pleiotropy

Carlos Cinelli Nathan LaPierre Brian Hill Sriram Sankararaman
Eleazar Eskin

August 30, 2021

Summary

We thank the reviewers for their comments and their time. In this response, we first provide a brief summary of the main points we have addressed. We then provide detailed answers to all reviewers' questions in the next two sections, one dedicated for each reviewer.

- **Additional simulations:** We performed all additional simulations and analyses as suggested by Reviewer 1. In summary we found that: (a) the simulations concentrating pleiotropic effects on a subset of variants reinforces the results of the previous simulation, and all methods still output excessively high false positive rates (we explain why below); (b) the simulation varying the relative strength of confounder led us to improve the presentation of the results, now focusing on how sensitivity analysis correctly describes to the user what she needs to believe in order to sustain a certain claim; and, (c) we performed additional sensitivity analysis regarding the effects HDL and LDL on coronary artery disease (CAD), and the effect of HDL is relatively less robust to residual pleiotropy.

- **Comparisons for the robustness value:** We now discuss the differences between our proposal and the approaches suggested by Reviewer 2. In brief, the E-value generalizes Cornfield conditions for risk ratios, whereas the robustness value generalizes Cornfield conditions for partial correlations and partial regression coefficients. Next, the “bias components” plots the reviewer suggests are not formal bounds on the strength of unobserved confounders, and if used as so, can lead users to erroneous conclusions even when observed variables are identical to unobserved variables, as we detail below. As requested by Reviewer 2, we also included MR-GENIUS in the simulations, which, as expected, has virtually 100% false positive rates in that setup. Finally, we explain how our method focuses on the sensitivity analysis of the MR *estimand*, and as such, it is compatible with any set of causal assumptions (eg, monotonicity), so long as the identifying functional is the same.

All changes we performed in the manuscript are highlighted in blue. We do not highlight the supplementary material, since everything in there is new. We now turn to the point-by-point response, along with our revisions, for each reviewer.

Detailed response to Reviewer 1

R1.1. *The authors propose an approach for quantifying the robustness of a Mendelian randomization analysis based on a polygenic risk score. This is an important issue, and the proposed method is elegant and produces convenient visualizations. I have some comments (below) but overall think this is really nice work.*

Response: We thank the reviewer, and really appreciate the positive comments.

R1.2. *Many confounders may explain a much larger proportion of variance than any measured confounder including PCs. This can happen for example when two traits have a high genetic correlation or when the PRS used as the instrument is constructed from a small number of variants, some of which have large pleiotropic effects. The authors do state that “Benchmarks for horizontal pleiotropy, on the other hand, require specific knowledge of the aetiology of the disease under study, or of the social process under investigation.” However, I think this issue is worth a little more discussion. In particular, I think that PCs are probably bad benchmarks for pleiotropic effects (though they are fine for population structure effects).*

Response: We thank the reviewer for the suggestion. As requested, we have added a little more discussion on the use of observed variables to aid judgments about the plausible strength of unobserved variables on the results section. In particular, we agree with the reviewer that PCs are better benchmarks for population structure, rather than pleiotropy. Our current guidelines for residual pleiotropy is to benchmark it against observed putative pleiotropic pathways guided by substantive knowledge. Although, due to data limitations, there may be ambiguity as to whether some of these variables could be colliders (as we discuss in the answer to **R1.6** below), the results of the sensitivity analysis can still be illuminating, and at a minimum act as useful warnings, especially when it overturns results.

R1.3. *Continuing the thought of point 1, this could be explored further in the real data example by looking at “messier” examples. One trait pair to consider would be HDL cholesterol and coronary artery disease. This is often considered an MR false positive (though there is some argument) that arises due to pleiotropy between HDL and other CAD risk factors including LDL, triglycerides, and BMI. I expect the pleiotropic confounding in this case would be much stronger than in the BMI -> townsend index case so it would be an interesting comparison.*

Response: We thank the reviewer for the suggestion. We have added the analysis of the effect HDL and LDL on coronary artery disease (CAD) in the supplementary material. Briefly, this is what we find. Our focus is on the sensitivity of the PRS-outcome association (i.e., assessing the null of zero causal effect). First, the robustness value and partial R^2 turned out to be quite similar for both HDL and LDL. However, the robustness to benchmarks of residual pleiotropy were indeed different. Interestingly, for instance, we found that the effect of HDL on CAD *would not* survive residual pleiotropy as strong as LDL+triglycerides, whereas the effect LDL on CAD *would* survive residual pleiotropy as strong as HDL+triglycerides. So in that sense, the effect of LDL is relatively more robust. We further note that the goal of sensitivity analysis here is not to give user a “context-free” answer, but rather to tell the user what one needs to know in order to maintain or refute a claim. It is now incumbent upon experts of this topic to judge whether pleiotropic channels with the strengths revealed to be problematic can be ruled out or not (i.e., whether residual pleiotropy with such strength is plausible). One final note for this particular example is that CAD is a binary outcome, and our

method targets the sensitivity of the traditional MR estimand, consisting of the ratio of two OLS coefficients. This may not be the best approach for such cases (although it does not matter much in this particular example). Extension of these sensitivity tools to non-linear models is currently under work.

R1.4. *This method applies only to MR using a single variant (probably a polygenic score), making it impossible to consider effect [heterogeneity]. Most methods that relax MR assumptions including MR-PRESSO, GSMR (Zhu et al, 2018), CAUSE (Morrison et al, 2020), and MR-MIX (Qi and Chatterjee, 2019) do so by assuming that a pleiotropic confounder will induce heterogeneity in the $\beta_{\text{hat_outcome_j}} / \beta_{\text{hat_exposure_j}}$ where j indexes the variant. This heterogeneity is then an independent piece of information to the data used in the proposed robustness metrics. I think it would be good to include a more nuanced discussion of this issue. Additionally, since robustness metrics cannot protect against pleiotropy that is stronger than existing benchmarks, would it be reasonable for investigators to deploy a hybrid approach looking at results from both types of methods?*

Response: We agree with the reviewer that sensitivity analysis should be seen as a complement to current practices, by quantifying the robustness of traditional MR to residual biases. We have included this explicitly in the discussion. We believe investigators should indeed deploy a hybrid approach and triangulate evidence from different sets of assumptions, and thus also perform alternative analyses such as MR-Egger, MR-Presso, MR-Mix, among others. As the reviewer mentions, these analyses also reveal extra pieces of information about the data, such as possible effect heterogeneity. As for the robustness metrics, we clarify that it can tell the researcher whether the results are robust to pleiotropy of any hypothetical size. Sometimes results can be robust to pleiotropy even multiple times stronger than existing benchmark covariates, such as the example of the effect of BMI on DBP. There, the effect is robust to pleiotropy up to 7 times stronger than observed putative pleiotropic pathways.

R1.5. *Following from above, in the simulations, pleiotropic effects are spread evenly over all 90 variants, meaning that there is no or little heterogeneity in the causal effects implied by each individual variant. In this scenario, MR- PRESSO should perform nearly identically to plain IVW so, as presented, those results are really “strawman” results. It would be nice to see additional simulations that a) concentrate pleiotropic effects on only a subset of variants and b) have a pleiotropic confounder that has stronger effects than the measured confounder. I appreciate that the authors point out that “The phenomenon demonstrated in the simulation is simply the well-known but often overlooked fact that, with large enough data, any residual bias will eventually be statistically significant.”*

Response: We thank the reviewer for the suggestion, we have performed all the requested additional simulations, namely, (a) concentrating pleiotropic effects on subset of the variants; and, (b) varying the relative strength of the unobserved pleiotropic trait, as compared to the observed one. These results are in the paper, but for convenience we summarize our findings here.

The simulations of (a) were especially illuminating. First, since MR-PRESSO requires the InSIDE condition to work, it continues to output virtually 100% of false positives even when we allow almost all variants to be valid. In fact, by revisiting the original MR-PRESSO paper, we see that the authors found similar results in their own simulations (Table 8 of their supplementary material). Furthermore, of special interest were the results of additional methods we included in the simulation, such as MR-MBE (Hartwig et al., 2017) and MR-Mix (Qi and Chatterjee, 2019). In theory, these methods should work as soon as the “plurality rule” is valid, meaning that the modal value of the pleiotropic effects is zero (known as the “ZEMPA”—Zero Mode Pleiotropy Assumption). Thus, if the

mode of variants is valid, ZEMPA holds. In practice, however, we still find excessively high false positive rates for all these methods. Since finite sample biases may still remain, default choices for the tuning parameters may not be adequate for such large sample sizes. This is in agreement with Hartwig et al. (2017) original simulation studies, the difference being, given their smaller sample size, this fact was not so pronounced in their case. Thus, we believe these additional simulations further reinforce the message of that section—with genetic databases reaching hundreds of thousands, if not millions of observations, any discrepancy may eventually be detected, even if small and spurious. Quantifying how strong residual biases need to be to explain away such findings is thus an important step to make sense of these results.

# of Valid Variants	Proportion of rejections of the null ($\alpha = 5\%$)						Critical k		$RV_{\alpha=0.05}$	
	IVW	PRESSO	Egger	GENIUS	MBE	Mix	5th	95th	5th	95th
20	100%	100%	99%	100%	88%	67%	0	0.9	0%	0.5%
40	100%	100%	99%	100%	91%	60%	0	0.9	0%	0.4%
60	99%	99%	98%	100%	87%	56%	0	0.9	0%	0.3%
80	57%	56%	68%	100%	55%	25%	0	0.7	0%	0.1%
90	4.3%	4.6%	3.5%	100%	0.5%	2.1%	0	0	0%	0

Table 1: Simulation in which pleiotropic pathways are concentrated on only a subset of variants. Sample size of 450,000 and $\tau = 0$. All methods are run with default parameter choices.

Regarding (b), the suggestion made us realize that the way the results were presented may have given the impression that the relative strength of $k = 1$ had some special status to it. Thus, first, we improved the presentation of the original results. We now show the “critical k ” (5th and 95th percentiles) that the analyst would have observed in her contour plots. Visually, the critical k is the maximum multiple of the relative strength k of the benchmarks until it reaches the critical line of interest (see also the new subsection in Methods). For example, in the DBP analysis, the critical k for a confounder k times as strong as the principal components was about 14; whereas, in the Townsend deprivation index analysis, this critical k was about 0.5. Thus the results show that, in over 95% of the simulations, the critical k is below 1, meaning that the sensitivity analysis would correctly reveal that a confounder as strong as X is able to overturn the results.

Next, we performed the additional simulations as requested. These are shown in Table 2 below, varying the relative strength of the confounder. Note how the sensitivity analysis correctly reveals that it would take a an unobserved confounder W about 1-2, 2-3, or 3-4 times stronger than X to make the MR estimate statistically insignificant (since we are considering statistical significance, the “sample” k is more conservative than the population k). In other words, the sensitivity analysis is telling the correct story to the user: either there exists some confounder about k times as strong as X , or, if such strengths are not possible (we know they are in the simulation), then there must be some true causal effect in order to explain the observed association.

Relative strength of W	Proportion of rejections of the null ($\alpha = 5\%$)						Critical k		$RV_{\alpha=0.05}$	
	IVW	PRESSO	Egger	GENIUS	MBE	Mix	5th	95th	5th	95th
2	100%	100%	86%	100%	100%	75%	1.1	1.9	1.0%	1.3%
3	100%	100%	99%	100%	100%	74%	2.0	2.8	1.5%	2.0%
4	100%	100%	100%	100%	100%	75%	2.8	3.9	2.2%	2.7%

Table 2: Simulation varying the strength of unobserved variable W relative to the observed variable X . Sample size of 450,000 and $\tau = 0$. All methods are run with default parameter choices.

R1.6. *It seems plausible that smoking and alcohol consumption are colliders in the BMI-> Townsend index analysis, i.e. it seems plausible that higher Townsend deprivation index causes increased drinking or smoking and that there are genetic variants in the PRS linked to drinking and smoking. This would induce bias in the estimate obtained.*

Response: We agree, this is an unfortunate practical problem with most traits in datasets with insufficient temporal information (such as the UKBB), and we can only fully solve it with better data. For instance, if the measured smoking and drinking behavior referred unambiguously to *past* behavior (i.e, behavior measured before deprivation), then we could be assured they are not colliders, since current deprivation cannot affect past behavior. However, in datasets like the UKBB, we do not have access such type of longitudinal data. Moreover, these variables can also plausibly act as pleiotropic pathways (or proxies of pleiotropic pathways), since genetic variants may affect behavior which also affects deprivation. Since there is not much of a solution to this, in practice our recommendation is as follows: if the researcher is 100% positive that these variables could *only* be introducing collider bias, then they should not be included. If, however, there is ambiguity as to whether they also act as (proxies of) pleiotropic pathways, which we argue to be the case here, then we recommend that one should still assess whether the results are sensitive to such variables, or unobserved variables as strong as such variables. We added this explanation on the results section.

R1.7. *I think it would be useful to explain the relationship between the partial R^2 and RV values and the contour plots, since the verbal description is somewhat hard to follow. Is it correct that the dotted red line should have a vertical asymptote at the partial R^2 for the PRS-Outcome relationship and that the point (RV_α, RV_α) should lie on the red line? (this doesn't look correct in the BMI-Townsend index case).*

Response: This is a great suggestion, these visualizations are indeed very useful. We have included them along with detailed explanations in the methods section, and we also point readers to that in the methods overview of the results. In short, the point RV_α, RV_α indeed lies in the red line (more precisely, it lies where the contour crosses the 45-degree line—but note in most plots the horizontal and vertical axis are *not* in the same scale; we do this to properly show the benchmarks). As for the partial R^2 , it is the vertical asymptote of the *zero* contour. The vertical line tangent to any contour for a given significance level α is given by the *extreme robustness value*, XRV_α , derived in Cinelli and Hazlett (2020b), which is given by:

$$XRV_\alpha = \begin{cases} 0, & \text{if } f_\alpha \leq 0 \\ \frac{f^2 - f_\alpha^{*2}}{1 + f^2}, & \text{otherwise.} \end{cases} \quad (1)$$

where $f^2 = \frac{R^2}{1-R^2}$ is Cohen’s partial f^2 statistic. We thus have that the RV is the point of equal association of the contour, and the XRV the line tangent to that contour, which is never crossed, as shown in Figure 1 (Cinelli, 2021, p. 69). Note when we set $\alpha = 1$ (that is, fully eliminating the point estimate, which corresponds to the zero contour) the XRV reduces to the partial R^2 , i.e, $XRV_{\alpha=1} = \frac{f^2}{1+f^2} = R_{Y|Z|X}^2$. For practical purposes in MR studies, we chose to give preference to the partial R^2 . This is because the partial R^2 is a well known metric among MR researchers, and it is also already common practice to report the partial R^2 of the genetic instrument with the exposure (to assess the problem of “weak” instruments). Therefore, instead of introducing yet another metric, we simply suggest additionally computing the partial R^2 for the genetic association with the outcome,

Figure 1: Contour plot illustrating the RV and XRV. The partial R^2 is the XRV for the zero contour line.

which comes naturally in this setting. Finally, note *all* sensitivity statistics are simply transformations of the partial R^2 (through Cohen’s f), thus by always reporting the partial R^2 one can easily transform this information to other sensitivity metrics.

R1.8. *It could be useful to also show the contour plot for partial R^2 PRS vs partial R^2 exposure for at least one of the BMI applications in order to illustrate the meaning of the summary stats for the PRS-Exposure relationship.*

Response: The sensitivity plot for the genetic association with the exposure (BMI) is now included in the main text, Figure 4.

R1.9. *The last line of eq (7) follows immediately from the last line of eq. (6) so the preceding derivation could be omitted.*

Response: We omitted most of the preceding derivation as suggested.

R1.10. *f_Z and f_Y appear in eq (15) but are not defined until the following paragraph. Should f_Z in the first part of eq (15) be R_Z ?*

Response: The equation is correct, it should be f_Z (partial Cohen’s f). This adjustment (using f_Z instead of the correlation R_Z) is necessary to formally bound the strength of the unobserved confounder. This happens due to the changes in the baseline variance to be explained by the confounder, and failing to account for that can lead to “optimistic” benchmarks. Further discussion can be found in Cinelli and Hazlett (2020a, Section 6.2). For clarity, we have also moved the definition of partial Cohen’s f to the main text, instead of leaving in it as a footnote.

Detailed response to Reviewer 2

R2.1. *The authors present methods and opportunities for bias and sensitivity analyses in MR. I greatly value the endeavor of the authors. However, I have two major themes of concerns about the current version. One, the authors have not engaged with the work on similar methods already appearing in epidemiology, and thus it is hard to gauge the novelty of the contributions here (see some questions below). Two, I think the framing of the paper should better acknowledge the scope of the settings and types of bias being studied, and I'm concerned that when this scope gets clarified then these tools may not actually be as helpful as presented in real applications with all the complexities of human health. My comments below reflect these two themes.*

Response: We thank the reviewer for raising such concerns, clarifying them made the paper stronger. In particular, in the sequence of responses below, we show: (i) how the RV is different from the E-value, and how it advances this rich tradition started by Cornfield et al. (1959) to the case of multivariate regression; (ii) how “bias components plots” are *not* formal sensitivity analysis to *unmeasured* confounders, and can lead users to erroneous conclusions if used for that purpose. This is why we need formal bounds such as the ones we present on the paper. And, finally, (iii) we clarify how our sensitivity analysis, by focusing on the IV estimand, is flexible enough to accommodate many different sets of causal assumptions, so long as the resulting estimand is the traditional IV estimand. Thus, we can handle different types of violations of the exclusion restriction, of instrument-outcome unmeasured confounding, or even selection bias, as we demonstrate in an example suggested by the reviewer in **R2.6**. In this sense, we argue that sensitivity analysis is an important tool to make MR more suited to handle the imperfections and complexities of human health.

R2.2. *How does your 'robustness value' and worst-case-bounds correspond to prior works on the E-value for MR (Swanson Vanderweele *Epid.*), bias component plots (Jackson Swanson *Epid.*; Davies *Epid.*), and other approaches applied or described previously (e.g., Glymour et al. *AJE* 2012; Vanderweele et al. *Epid.*; the chapter on IVs in *Modern Epidemiology's* latest edition)? You may also want to compare your useful Figures with those that are common in pharmacoepidemiology settings, not for IVs necessarily but for quantifying confounding bias; and the original and canonical epi papers on this kind of topic from 70+ years ago by Cornfield, Bross, etc. (again, not in the setting of IVs).*

Response: We thank the reviewer for mentioning these related works, we now cite and discuss them in the paper. We note the robustness value (not in the context of MR) was first introduced in Cinelli and Hazlett (2020a), and many of these discussions can also be found there. For convenience, here we reproduce the relevant parts and add additional remarks that addresses the specific references raised by the reviewer.

- **Cornfield:** The partial R^2 and the robustness value (RV) of Cinelli and Hazlett (2020a) can be seen as following the tradition of Cornfield et al. (1959) in deriving conditions that the confounder must meet in order to logically account for an observed association. The difference is that the partial R^2 and the RV are generalizing “Cornfield conditions” for partial regression coefficients, not for risk ratios. This also makes it easier to understand the difference between the RV and the E-value, since the E-value focuses on generalizing the “Cornfield conditions” for risk-ratios, as we discuss below.

- **E-value:** Regarding the E-value of VanderWeele and Ding (2017), Cinelli and Hazlett (2020a) already discussed some of the differences between them (page 61):

As to VanderWeele and Ding (2017), the authors have recently advanced the E-value, a sensitivity measure suited specifically for the risk ratio. For other effect measures, such as risk differences, the E-value is an approximation, whereas if the researcher uses linear regression to obtain an estimate, the robustness value is exact. Also, while the robustness value parameterizes the association of the confounder with the treatment and the outcome in terms of percentage of variance explained (the partial R^2), the E-value parameterizes these in terms of risk ratios. Whether one scale is preferable over the other depends on context, and researchers should be aware of both options.

Now moving to the specific context of Mendelian randomization, note that effect measures such as the partial R^2 are ubiquitous in genetics (e.g, see notions such as “heritability”). In this sense, the robustness value and the partial R^2 seem natural for these settings. To illustrate, in our running example, one obtains the E-value *approximations* for the standardized linear regression coefficients of 1.15 (1.13 for the 95% CI) for DBP, 1.07 (1.03 for the 95% CI) for Deprivation, and 1.50 (1.49 for the 95% CI) for the first stage (PRS-BMI association). If we take some rules-of-thumb suggested by the literature (VanderWeele and Ding, 2017; Swanson and VanderWeele, 2020) to judge what is a “big” or “small” E-value, we would conclude that “modest confounding” would be sufficient to explain away the genetic associations with BMI or DBP. Our analysis, however, reveals that it does take some fairly strong confounding to do so. The problem here is simply that the risk ratio scale is not natural in this setting. Thus, in this example, the partial R^2 parameterization has the benefit of speaking directly to quantities researchers in these areas are used to thinking about (as another example of how these metrics are pervasive, note that many simulations in MR parameterize the strength of genetic effects in terms of partial R^2). But we would like to reinforce the original statement of Cinelli and Hazlett (2020a)—researchers should be aware of both options, and, depending on context, one metric can be easier to reason about than the other. Finally, another difference between the two methods is that, under the E-value framework, currently there are no formal benchmarking procedures to leverage observed covariates to bound the strength of the unobserved confounder. Recent attempts to fill this gap (D’Agostino McGowan and Greevy Jr, 2020) fall into what is known as “informal benchmarking” (Cinelli and Hazlett, 2020a, Sections 4.4 and 6.2), and may lead users to erroneous conclusions. In fact, this is a problem that can also affect bias components plots whenever they are used to extrapolate claims about *unmeasured* confounders, as we discuss next.

- Bias components plots: The bias components plots suggested by Jackson and Swanson (2015) and Davies (2015) can be useful to understand and decompose the difference between an IV estimate including and excluding *observed* variables, and contrast this to usual “backdoor adjustment” estimates, again including and excluding *observed* variables. These tools, however, do not provide formal sensitivity analysis due to unmeasured confounding, as we do here. Moreover, while Jackson and Swanson (2015, p. 503) claim that bias components plots are “*informative to potential bias when the type of unmeasured confounding is expected to be similar to what we observe,*” this claim is not necessarily correct either, even in the ideal case of having unmeasured confounder that is *identical* to the measured confounder in its associations with all other variables of the system. To witness, consider the model below (interpret this as pseudocode to create the data, “expit” is the inverse of the logit function):

$$W \leftarrow \text{Binomial}(p = 0.5) \tag{2}$$

$$X \leftarrow \text{Binomial}(p = 0.5) \tag{3}$$

$$Z \leftarrow \text{Binomial}(p = \text{expit}(W + X)) \tag{4}$$

$$D \leftarrow \text{Binomial}(p = \text{expit}(Z + W + X)) \tag{5}$$

$$Y \leftarrow \text{Normal}(\mu = X + W, \sigma^2 = 1) \tag{6}$$

Where, as in the paper, Z is the instrument, D the exposure, Y the outcome, and X and W are the confounders. Note how the observed confounder X and the unobserved confounder W are *completely symmetric*. Also, note the true causal effect of D on Y is zero. Therefore, we would like the result of a sensitivity analysis to tell us that an unmeasured variable W “as strong as” the measured variable X would be sufficient to overturn the observed results. Yet, if one uses the “observed bias” due to X to infer the bias due to the unmeasured variable W (without proper adjustment) this leads to the incorrect conclusion that an “unmeasured confounder W similar to the measured confounder X ” would not be sufficiently strong to explain away the IV estimate, when in fact it would. The main problem here is collider bias; although X and W are independent, they become dependent after conditioning either on D or on Z , and thus the observed association of X with Y is dampened. For this reason we recommend using formal bounds on the unmeasured confounder, such as the ones we discuss in the paper. Further discussion of this problem can be found in Cinelli and Hazlett (2020a, Sections 4.4 and 6.2).

- **Other cited references:** We read all the remaining references suggested by the reviewer, namely, Glymour et al. (2012), VanderWeele et al. (2014), and the chapter on IVs of Lash et al. (2020). The first reference (Glymour et al., 2012) does not directly offer a framework for sensitivity analysis (though it does mention that sensitivity analysis is important), rather it discusses testable implications of MR when one imposes additional constraints on the model (see our discussion of testable implications in the answer to **R2.4** below). The main focus of the second reference (VanderWeele et al., 2014) is to illustrate different ways in which violations of the IV assumptions can happen. As we explain in detail in the answers to **R2.5**, **R2.6** and **R2.7** below, our sensitivity results hold regardless of how violations happen, so long as the target of inference is the IV estimand, and thus it can cover the violations discussed in VanderWeele et al. (2014). Finally, chapter 3 of Modern Epidemiology (Lash et al., 2020) has a short discussion in which it mentions the possibility of performing sensitivity analysis and using the E-value, which we have addressed above.

R2.3. *You state boldly (as in you write it in bold) that current proposals fail whereas there toolbox does not. This again needs to acknowledge better the scope of the other proposals you are comparing to. It doesn't seem to reference the methods above, nor does it reference some of the novel 'robust' approaches like MR-GENIUS (Tchetgen Tchetgen et al.). It seems like you are primarily concerned in comparing to the popular MR-Egger. Though this limited comparison is illuminating given MR-Egger's popularity, it is not really engaging with the rich methods literature here and thus not advancing our collective understanding of how to do MR well with all tools available.*

Response: We thank the reviewer for suggesting other related work for us to assess, and we have expanded our simulations accordingly. First, we note that the methods previously mentioned are already addressed in the answers to **R2.1** and **R2.2**. Regarding MR-GENIUS, as requested by the reviewer, we have added it to the simulations (along with two new methods, namely, MR-MBE and MR-Mix). The updated tables can be found in the text, but are also provided here for convenience, in the response to **R1.5**. Since the simple DGP of the simulation study does not satisfy MR-GENIUS required assumptions (e.g, GENIUS requires heteroskedastic error terms for the first stage), MR-GENIUS fails to control the type-I error rate, as expected. Thus, in the supplementary material, we included additional simulations in which we introduce heteroskedasticity just to comply with MR-GENIUS assumptions. We then investigate how it behaves under various strengths of heteroskedasticity. We find that MR-GENIUS still outputs excessively high false positive rates when the level of heteroskedasticity is low. For convenience, the results are reproduced here, in Table 3.

	Strength of heteroskedacity σ						
	0	0.25	0.5	0.75	1	2	3
False Positives	100%	99.9%	89.7%	51.6%	26.7%	7.5%	5.6%

Table 3: False positives for MR-GENIUS when varying the strength of heteroskedasticity as parameterized by σ . Sample size fixed at $N = 100,000$, true causal effect set at $\tau = 0$.

R2.4. *Can you comment on whether conclusions drawn from these approaches are always consistent with the observable data? Would applied researchers need to also consider the IV inequalities or other approaches (e.g., Balke and Pearl; Bonet; Diemer et al.), either prior to implementing all analyses or even integrating them somehow into your approaches?*

Response: This is a rather technical topic. Briefly, the answers are: (i) Sensitivity parameters may indeed be constrained by the observed data, and this applies to *any* sensitivity analysis (such as the E-value). However, characterizing the feasible region in the standard setting may not have much practical value. (ii) Whenever applicable, researchers should still check the IV inequalities, and this *does not* preclude sensitivity analyses.

We now elaborate in detail. First note that testing the model against the data and performing sensitivity analysis should be seen as distinct (although complementary) tasks. More specifically, the former is asking whether the data is compatible with the assumptions of the model, while the later is asking whether the study’s conclusion is sensitive to plausible violations of these assumptions. That said, and before moving to instrumental inequalities, the observed data may indeed constrain the region of sensitivity parameters that are logically possible. This may happen to *any* sensitivity analysis, unless it was constructed specifically to have no testable implications (such as in Franks et al. (2019)). For instance, the sensitivity parameters related to the E-value suggested by the reviewer (Ding and VanderWeele, 2016; VanderWeele and Ding, 2017; Swanson and VanderWeele, 2020), may have a constrained feasible region, and thus not all values may be consistent with the observed data (see for instance, recent discussion by Sjölander (2020) and Peña (2021)). More concretely for our case, since we are dealing with partial correlations (R^2), one such immediate constraint is that the observed covariance matrix combined with the sensitivity parameters must imply a valid covariance matrix (i.e, it must be positive semi-definite). Although this might rule out certain combinations of sensitivity parameters, the bias is still unconstrained (Zheng et al., 2021). For practical purposes, ignoring the feasible region of the sensitivity parameter is simply conservative, meaning that accounting for it can only, if anything, make results *less* sensitive.

Now moving specifically to the instrumental inequalities (Pearl, 1995; Swanson et al., 2018; Kédagni and Mourifié, 2020). First, recall that if the exposure trait is continuous, then the IV model entails no testable implications without further structural assumptions (Bonet, 2001; Gunsilius, 2020). Therefore, in that setting, the IV inequalities should not impose further constraints on the feasibility region of the sensitivity parameters. Considering now the cases where the instrumental inequalities may apply, they can be related to the sensitivity analysis of the MR estimate in at least two ways. First, they can act as useful warnings. That is, if the data is not compatible with the instrumental inequalities, the test immediately warns the researcher of the need to conduct a sensitivity analysis, since we now know the MR assumptions must be false. However, we remind readers that “passing” the instrumental inequality test does not preclude the need for a sensitivity analysis—just because the data is compatible with the MR assumptions, this does not mean they are true. Second, more technically, it may be the case that instrumental inequalities constraint the feasible region of the joint set of sensitivity parameters, meaning that perhaps not all combinations

of sensitivity parameters are logically valid. We are not aware of any work on this area, and such considerations also hold for *any* sensitivity for MR, including the ones suggested by the reviewer, such as the E-value proposal of Swanson and VanderWeele (2020). For the reasons articulated in the previous paragraph, characterizing the exact feasible region seems to have low practical value when dealing with one instrument (such as a polygenic risk score) and one treatment, which is our focus in this paper. So we do not see much benefit in pursuing this here. When extending these results to handle multiple instruments and multiple treatments, however, the constrained feasible region may actually be very informative, as recent work by Zheng et al. (2021) suggests. We leave this as an interesting direction for future work.

R2.5. *Can you comment on how much each presented approach relies on assumptions of homogeneity, and if so what specific assumptions? Are these reasonable assumptions to be making in real MR studies? If yes, why? If no, how does that affect interpretability of these checks?*

Response: Our approach does not rely on the assumption of homogeneity. We now state this explicitly in the text. To understand why, note our focus is on the sensitivity analysis of the IV estimand, namely, the ratio:

$$\tau := \frac{\beta_{YZ|\mathbf{X}\mathbf{W}}}{\beta_{DZ|\mathbf{X}\mathbf{W}}} \quad (7)$$

This IV estimand can have different causal interpretations, depending on extra assumptions the researcher is willing to defend (Swanson et al., 2018; Angrist and Pischke, 2009). For instance, under certain assumptions of effect homogeneity, τ can be interpreted as the average treatment effect. In the binary setting, under the assumption of monotonicity, τ can be interpreted as a local average treatment effect. Moreover, if one is interested in testing the sharp null hypothesis of zero effect for all individuals, effect homogeneity holds by definition (in all scales) and thus it is not a relevant concern. In sum, we focus on the IV estimand precisely to provide this type of flexibility. So long as the researcher’s target is the IV estimand above, all of our sensitivity tools can be used, regardless of the particular set of causal assumptions that justified the target of inference.

R2.6. *It needs to be clearer to the end user what specific DAGs might be valuable to consider here, as the same IV assumptions can be violated due to many different structures of violations (see Vanderweele et al. Epid. for some examples of the exclusion restriction; see Swanson Epid. for some examples of collider biases; just for a few examples). If these sensitivity analyses ‘work’ for thinking about multiple these types of biases, that should be clear; if they do not, the scope should be clear. Along the same lines, does this all work regardless of if Z, D, Y, X are continuous or dichotomous and whether Z is the risk score as described here vs. a vector of separate proposed instruments? This all is also important for contextualizing your findings against existing methods.*

Response: We thank the reviewer for suggesting various DAG structures for us to consider. The answer is analogous to the response to **R2.5** above. Our sensitivity analysis is compatible with *any* DAG in which Z is a valid instrument for the causal effect of D on Y conditional on observed covariates \mathbf{X} and hypothetical unobserved covariates \mathbf{W} , so long as the identifying estimand is the traditional IV estimand of Equation 7. Therefore, although violations could arise due to very different structures, the mechanics of the sensitivity analyses remains the same. For concreteness, we demonstrate this in some examples of the references suggested by the reviewer (VanderWeele et al., 2014; Swanson, 2019). These examples are also included in the methods section.

Figure 2: Different IV models with alternative substantive interpretations, yet resulting in the same procedure for sensitivity analysis.

Figure 2 illustrates some cases found in VanderWeele et al. (2014):

- Figure 2a: suppose we have a time-varying exposure D_1 and D_2 that affects Y . If the researcher is interested in the causal effect of D_2 on Y , the conditional set $W = \{D_1\}$ is sufficient for making Z a valid genetic instrument. Thus, if D_1 is not measured, the sensitivity parameters here would consist of postulating hypothetical strengths for how much residual variation Z explains of D_1 , and how much residual variation D_1 explains of Y .
- Figure 2b: suppose we are interested in the causal effect of D on Y , but the genetic instrument Z also affects trait W , which is not measured. Here W is again sufficient to make Z a valid instrument, and the sensitivity parameters would have the same interpretation as before.
- Figure 2c: suppose we are interested in the causal effect of D on Y , but that the genetic instrument Z_1 is in linkage disequilibrium with another variant Z_2 . As before, if Z_2 is not measured, one can perform sensitivity analysis by positing hypothetical strengths of the linkage disequilibrium (as parameterized by the partial correlation of Z_1 and Z_2), and how much residual variation the genetic variant Z_2 explains of Y .

Basically all examples in VanderWeele et al. (2014) are variations of these structures, and one can consider a set of hypothetical unobserved variables \mathbf{W} that, if conditioned upon, would render Z a valid genetic instrument. One interesting exception is an example with measurement error in the exposure trait (Figure 6 of VanderWeele et al. (2014)). However, although one cannot estimate the causal effect unbiasedly in that example, one can still test the null hypothesis of *zero* effect—thus our sensitivity analysis for the zero null hypothesis still applies in that setting.

Now let us move to Figure 3, a case of “time zero misalignment” (Swanson, 2019). Note in this example we have selection bias, as both the genetic instrument Z and the unobserved variable W affect the indicator of whether the participant is selected to the study, S . The square node represents the fact that we always condition upon the variable S (and thus the colliding path $Z \rightarrow S \leftarrow W \rightarrow Y$ is open). Therefore, Z is not a valid IV for the study sample, $S = 1$. But further note that Z becomes a valid instrument for the study sample when we *further* condition on the unobserved variable W .

Figure 3: Z is not a valid instrument for the causal effect of D on Y for the selected sample $S = 1$. However, Z would be a valid instrument if we further adjust for W , and thus a sensitivity analysis can still be performed by hypothesizing different strengths for W , as in the main paper.

This is precisely the case of our setup, and thus one can use all of our tools for performing sensitivity analysis in this setting. In particular, we can quantify how strong W must be in order to fully eliminate the genetic association with the outcome.

In sum, there are many ways in which the violations of IV assumptions can happen, and as likely as many ways one could posit hypothetical variables \mathbf{W} that would then render Z a valid genetic instrument. By design, our sensitivity analysis is agnostic to which specific causal assumptions one makes, in order to make it flexible enough to accommodate any of these models. The only requirement is that the researcher has decided her target of inference is the estimand of Equation 7.

R2.6. *MR inherently is studying time-varying exposures with proposed IVs fixed at conception, yet 'the' DAG drawn by everybody is of a time-fixed or point exposure. How do your methods address this? What effect are you even assuming to be estimating - e.g., a lifetime effect (Labrecque AJE) or something else? Or simply a test of a sharp causal null hypothesis (Swanson et al. EJE)? I think the answer is just the test, given what you write near the beginning of page 3, but even so it is not clear if your approach contextualizes only a test for a sharp null of a point exposure and not a joint sharp null. In short, it seems like trying to understand bias means you first need to make the causal question clear, and that may mean acknowledging that this canonical IV DAG is not the best starting point for MR. Let me just acknowledge in raising this specific point, however, that nearly all MR methods and applications are fuzzy about this, so this is not unique to the current paper..*

Response: We thank the reviewer for raising this point. We acknowledge this is a complicated issue and that most MR applications fail to properly define the target causal effect which they aim to estimate. However, from a methodological point of view, the answer to this question is similar to the previous answers. Our sensitivity analysis is agnostic with respect to the specific causal query one wishes to estimate, so long as the resulting *statistical estimand* is given by the traditional IV estimand of Equation 7. For concreteness, we use an example of Labrecque and Swanson (2019) to illustrate why this is the case.

Figure 4 replicates Figure 3 of Labrecque and Swanson (2019), with the addition of hypothetical unobserved variables W_1 and W_2 that could block pleiotropic pathways (there are many different ways one could hypothesize them, this is just one illustration), and bidirected arrows explicitly representing unmeasured confounders of the exposures D_1 and D_2 and the outcome Y . For simplicity, we also consider a linear structural model, as in Labrecque and Swanson (2019).

Figure 4: Different sensitivity analysis can be performed depending on the target effect of interest.

Thus let us start by considering that the researcher is interested in the “contemporaneous” causal effect of D_2 on Y , namely, the path coefficient λ_{d_2y} . In this model this can be obtained using Z as an IV with the conditioning set $\{W_1\}$. Now suppose the investigator is interested in the “long term” causal effect of D_1 on Y (namely, the total effect $\lambda_{d_1y} + \lambda_{d_1d_2} \times \lambda_{d_2y}$). Here the conditioning set

$\{W_2\}$ would be sufficient to render Z a valid instrument for this task. Finally, suppose interest lies in the effect of a joint intervention on D_1 and D_2 (this amounts to the sum $\lambda_{d_1y} + \lambda_{d_2y}$). The second component of the sum has already been solved, and the first component can be estimated using Z as an instrument with the conditioning set $\{W_2, D_2\}$. In sum, for all these cases, sensitivity analysis would proceed as before, by postulating the maximum explanatory power of the relevant unobserved variables \mathbf{W} with Z and the relevant trait, possibly after partialling out a set of observed variables. Note what changes is the interpretation, not the procedure. This is why we do not specialize the tools for a particular query, but instead we focus on the sensitivity of the IV estimand.

References

- Angrist, J. and Pischke, J.-S. (2009). *Mostly harmless econometrics: an empiricists guide*. Princeton: Princeton University Press.
- Bonet, B. (2001). Instrumentality tests revisited. In *Proceedings of the Seventeenth conference on Uncertainty in artificial intelligence*, pages 48–55. Morgan Kaufmann Publishers Inc.
- Cinelli, C. and Hazlett, C. (2020a). Making sense of sensitivity: extending omitted variable bias. *Journal of the Royal Statistical Society: Series B*, 82(1):39–67.
- Cinelli, C. and Hazlett, C. (2020b). An omitted variable bias framework for sensitivity analysis of instrumental variables. *Working Paper*.
- Cinelli, C. L. K. (2021). *Transparent and Robust Causal Inferences in the Social and Health Sciences*. PhD thesis, UCLA.
- Cornfield, J., Haenszel, W., Hammond, E. C., Lilienfeld, A. M., Shimkin, M. B., and Wynder, E. L. (1959). Smoking and lung cancer: recent evidence and a discussion of some questions. *Journal of the National Cancer institute*, 22(1):173–203.
- D’Agostino McGowan, L. and Greevy Jr, R. A. (2020). Contextualizing e-values for interpretable sensitivity to unmeasured confounding analyses. *arXiv e-prints*, pages arXiv–2011.
- Davies, N. M. (2015). Commentary: an even clearer portrait of bias in observational studies? *Epidemiology (Cambridge, Mass.)*, 26(4):505.
- Ding, P. and VanderWeele, T. J. (2016). Sensitivity analysis without assumptions. *Epidemiology (Cambridge, Mass.)*, 27(3):368.
- Franks, A., D’Amour, A., and Feller, A. (2019). Flexible sensitivity analysis for observational studies without observable implications. *Journal of the American Statistical Association*, (just-accepted):1–38.
- Glymour, M. M., Tchetgen Tchetgen, E. J., and Robins, J. M. (2012). Credible mendelian randomization studies: approaches for evaluating the instrumental variable assumptions. *American journal of epidemiology*, 175(4):332–339.
- Gunsilius, F. (2020). Nontestability of instrument validity under continuous treatments. *Biometrika*.
- Hartwig, F. P., Davey Smith, G., and Bowden, J. (2017). Robust inference in summary data mendelian randomization via the zero modal pleiotropy assumption. *International journal of epidemiology*, 46(6):1985–1998.

- Jackson, J. W. and Swanson, S. A. (2015). Toward a clearer portrayal of confounding bias in instrumental variable applications. *Epidemiology*, 26(4):498.
- Kédagni, D. and Mourifié, I. (2020). Generalized instrumental inequalities: testing the instrumental variable independence assumption. *Biometrika*.
- Labrecque, J. A. and Swanson, S. A. (2019). Interpretation and potential biases of mendelian randomization estimates with time-varying exposures. *American journal of epidemiology*, 188(1):231–238.
- Lash, T. L., VanderWeele, T. J., Haneuse, S., and Rothman, K. (2020). *Modern Epidemiology*. Lippincott Williams & Wilkins.
- Pearl, J. (1995). On the testability of causal models with latent and instrumental variables. In *Proceedings of the Eleventh conference on Uncertainty in artificial intelligence*, pages 435–443. Morgan Kaufmann Publishers Inc.
- Peña, J. M. (2021). Simple yet sharp sensitivity analysis for unmeasured confounding. *arXiv preprint arXiv:2104.13020*.
- Qi, G. and Chatterjee, N. (2019). Mendelian randomization analysis using mixture models for robust and efficient estimation of causal effects. *Nature communications*, 10(1):1–10.
- Sjölander, A. (2020). A note on a sensitivity analysis for unmeasured confounding, and the related e-value. *Journal of Causal Inference*, 8(1):229–248.
- Swanson, S. A. (2019). A practical guide to selection bias in instrumental variable analyses. *Epidemiology*, 30(3):345–349.
- Swanson, S. A., Hernán, M. A., Miller, M., Robins, J. M., and Richardson, T. S. (2018). Partial identification of the average treatment effect using instrumental variables: review of methods for binary instruments, treatments, and outcomes. *Journal of the American Statistical Association*, 113(522):933–947.
- Swanson, S. A. and VanderWeele, T. J. (2020). E-values for mendelian randomization. *Epidemiology*, 31(3):e23–e24.
- VanderWeele, T. J. and Ding, P. (2017). Sensitivity analysis in observational research: introducing the e-value. *Annals of internal medicine*, 167(4):268–274.
- VanderWeele, T. J., Tchetgen, E. J. T., Cornelis, M., and Kraft, P. (2014). Methodological challenges in mendelian randomization. *Epidemiology*, 25(3):427.
- Zheng, J., D’Amour, A., and Franks, A. (2021). Copula-based sensitivity analysis for multi-treatment causal inference with unobserved confounding. *arXiv preprint arXiv:2102.09412*.

REVIEWERS' COMMENTS

Reviewer #1 (Remarks to the Author):

I appreciate the work the authors have put into this thorough and thoughtful revision. All of my comments are addressed satisfactorily. I think this work is an important and interesting addition to the MR literature.

Reviewer #2 (Remarks to the Author):

Thank you for engaging with the previous comments so thoroughly. My remaining concern is in the bold statements about how this method can generalize to thinking about any bias with respect to any causal estimand that might be estimated using the standard IV ratio. Bias is with respect to a causal question, so I cannot see from what they present that this would be true? Please expand more on the issue of the IV ratio being used to estimate a "complier" average causal effect, an average causal effect of a point exposure, an average causal effect of a period exposure, and a lifetime average causal effect, and why these formulas work for all. The authors do a nice job explaining why different bias structures can all be compatible with this, but do not formalize why the structures of bias all map onto the same (?) magnitude of bias for each possible question one could ask.

Intuitively, it makes little sense to me. If there is a backdoor path that is unblocked, and for whatever reason that structural bias operates only within the "never-takers", then there should be no bias in the CACE even if there is bias in the ACE. I sense a similar issue with the lifetime vs. point/period effect could also crop up.

Response to Reviewers—Nature Communications

Robust Mendelian randomization in the presence of residual population stratification, batch effects and horizontal pleiotropy

Carlos Cinelli Nathan LaPierre Brian Hill Sriram Sankararaman
Eleazar Eskin

November 3, 2021

We thank again the reviewers for their comments and their time. Responses to the remaining clarification questions are provided below.

Detailed response to Reviewer 1 (R1)

R1.1. *I appreciate the work the authors have put into this thorough and thoughtful revision. All of my comments are addressed satisfactorily. I think this work is an important and interesting addition to the MR literature.*

Response: We are really happy to hear that we have satisfactorily addressed R1's comments, and we thank the reviewer for the positive feedback.

Detailed response to Reviewer 2 (R2)

R2.1. *Thank you for engaging with the previous comments so thoroughly. My remaining concern is in the bold statements about how this method can generalize to thinking about any bias with respect to any causal estimand that might be estimated using the standard IV ratio. Bias is with respect to a causal question, so I cannot see from what they present that this would be true? Please expand more on the issue of the IV ratio being used to estimate a "complier" average causal effect, an average causal effect of a point exposure, an average causal effect of a period exposure, and a lifetime average causal effect, and why these formulas work for all. The authors do a nice job explaining why different bias structures can all be compatible with this, but do not formalize why the structures of bias all map onto the same (?) magnitude of bias for each possible question one could ask.*

Intuitively, it makes little sense to me. If there is a backdoor path that is unblocked, and for whatever reason that structural bias operates only within the "never-takers", then there should be no bias in the CACE even if there is bias in the ACE. I sense a similar issue with the lifetime vs. point/period effect could also crop up.

Response: We thank the reviewer for raising this question, this is an important point for clarification. The short answer is that, although we are using the same set of analytical *tools* for assessing these biases, as the reviewer correctly points out, the magnitude of the biases will not be the same when

Figure 1: Time-varying treatment example.

contemplating different target causal quantities. This happens because the set of (hypothetical) unobserved variables that should be part of the conditioning set will differ for each target causal effect of interest. We have included this explanation in the supplementary material.

To illustrate, let us consider again the case of a time-varying treatment, as in Figure 1. Here the model is simplified to focus on the essence of the question, so imagine we have linear structural equations and population data. Suppose we have one genetic instrument Z , which is known to affect some exposure trait D , and this exposure is suspected to have some effect on an outcome trait Y . The exposure, however, varies with time, so in fact “ D ” here really consists of D_1 and D_2 . Moreover, D_1 and D_2 were *not* measured. In this setup, our observed data is then simply the genetic association with the outcome, namely, the regression coefficient $\beta_{YZ} > 0$.

Now suppose we have two investigators: (i) investigator 1 is interested in the sharp null hypothesis that the treatment does not affect Y at all, at any time point; (ii) investigator 2 is interested in the null hypothesis that the treatment does not affect Y “contemporaneously” (i.e, the effect of D_2 on Y). As we have shown in the paper, in such cases, one needs only to worry about the sensitivity of the genetic association with the outcome β_{YZ} , and contemplate how strong an *appropriate* set of unmeasured variables \mathbf{W} needs to be in order to make $\beta_{YZ|\mathbf{W}} = 0$. So what are the sensitivity analysis results for each investigator?

Suppose we *knew* there are no further concerns of population structure nor pleiotropy, and Figure 1 reflects the true DGP. For our first investigator, we then trivially have that the appropriate conditioning set for her research question is $\mathbf{W} = \{\emptyset\}$. Thus, in scenario 1, our hypothetical \mathbf{W} explains zero residual variation both of the instrument and of the outcome, and a sensitivity analysis would trivially conclude that $\beta_{YZ|\mathbf{W}} = \beta_{YZ}$ —we thus still reject the null hypothesis. For our second investigator, on the other hand, the appropriate conditioning set is $\mathbf{W} = \{D_1\}$, and in general $\beta_{YZ|\mathbf{W}} = \beta_{YZ|D_1} \neq \beta_{YZ}$. Since D_1 was not measured, we have to contemplate how much residual variation Z could explain of the D_1 (which acts as a pleiotropic effect for D_2), as well as how much residual variation D_1 could explain of Y . Thus, depending on such plausibility judgments, we could perhaps conclude that it is unlikely that there is a “contemporaneous,” direct effect of D_2 on Y .

In conclusion, here we have the same data ($\beta_{YZ} > 0$), the same set of sensitivity analysis tools (i.e, contemplating how strong a hypothetical set of variables \mathbf{W} needs to be to bring the regression coefficient $\beta_{YZ|\mathbf{W}}$ to zero), yet, since the target causal effect of interest is different in each case, our conclusions and judgments about the magnitude of biases will also be different, as the reviewer correctly expects (because the hypothetical adjustment set \mathbf{W} is different in each case). Also note we could have made both scenarios more complex, by adding further confounders or pleiotropic effects for D_1 , for instance, but the essence of the example remains unaltered. We hope this clarifies the question.